# Mothers with higher twinning propensity had lower fertility in pre-industrial Europe

Ian J. Rickard[1,2], Colin Vullioud[2], François Rousset [3], Erik Postma [4], Samuli Helle [5], Virpi Lummaa [6], Ritva Kylli [7], Jenni E. Pettay[5], Eivin Røskaft [8], Gine R. Skjærvø [8], Charlotte Störmer[9], Eckart Voland [9], Dominique Waldvogel[10] & Alexandre Courtiol [2✉]

Historically, mothers producing twins gave birth, on average, more often than non-twinners. This observation has been interpreted as twinners having higher intrinsic fertility – a tendency to conceive easily irrespective of age and other factors – which has shaped both hypotheses about why twinning persists and varies across populations, and the design of medical studies on female fertility. Here we show in >20k pre-industrial European mothers that this interpretation results from an ecological fallacy: twinners had more births not due to higher intrinsic fertility, but because mothers that gave birth more accumulated more opportunities to produce twins. Controlling for variation in the exposure to the risk of twinning reveals that mothers with higher twinning propensity – a physiological predisposition to producing twins – had fewer births, and when twin mortality was high, fewer offspring reaching adulthood. Twinning rates may thus be driven by variation in its mortality costs, rather than variation in intrinsic fertility.

[1] Department of Anthropology, Durham University, Durham, UK. [2] Department of Evolutionary Genetics, Leibniz Institute for Zoo and Wildlife Research, Berlin, Germany. [3] Institut des Sciences de l'Évolution (ISEM), Université de Montpellier, CNRS, EPHE, IRD, Montpellier, France. [4] Center for Ecology and Conservation, University of Exeter, Penryn, UK. [5] Department of Social Research, University of Turku, Turku, Finland. [6] Department of Biology, University of Turku, Turku, Finland. [7] Department of History, University of Oulu, Oulu, Finland. [8] Department of Biology, Norwegian University of Science and Technology, Trondheim, Norway. [9] Institute for Philosophy, Justus Liebig University Gießen, Giessen, Germany. [10] Department of Evolutionary Biology and Environmental Studies, University of Zurich, Zurich, Switzerland. ✉email: courtiol@izw-berlin.de

Human twin births are rare but ubiquitous[1,2]. Despite a substantial research effort, evolutionary biologists[3–11], as well as developmental biologists[12], and geneticists[13,14], continue to struggle to understand why twinning occurs and what influences its prevalence. Twinning also represents a practical challenge for public health since, even in the presence of high-quality medical care, twin births increase hospital costs and mortality rates for both mothers and offspring across populations[15–18]. Here, we investigate what drives variation in twinning rates by examining how dizygotic twinning relates to maternal fertility.

The biology of twinning differs between monozygotic and dizygotic events[12]. Monozygotic twinning, which produces "identical twins", occurs at a low and steady rate across populations (~0.35–0.4% of births)[1] and results from the split of a fertilised embryo at an early developmental stage with no apparent reason or clear correlates (beyond being a frequent consequence of assisted reproductive technologies[19]). In contrast, dizygotic twinning, which produces "fraternal twins", results from the fertilisation of two eggs by two sperm and occurs at a rate that is higher and more variable both within and between populations (~0.7–2.7% of births)[1,2,20]. The rate of dizygotic twinning has been shown to be correlated with several environmental, genetic, and developmental factors, with a particularly strong effect of the age of the mother[13,20]. For these reasons, most research with a focus on variation in twinning rates focuses on dizygotic twinning, which we will henceforth refer to as *twinning* for short.

An idea well-entrenched in both the medical and evolutionary literature is that twinning may be the expression of a woman's *intrinsic fertility*—her potential to give birth irrespective of age and any stochastic factors occurring within her reproductive life, including past reproduction. Specifically, *twinning propensity*— the probability that a birth produces more than one offspring—is thought to be physiologically associated with higher intrinsic fertility[5–10,21]. Several authors have further proposed that, compared to mothers that have produced singletons only (*non-twinners*), the presumed higher intrinsic fertility of mothers that have produced twins (*twinners*) reflects their higher phenotypic quality—an idea known as the *heterogeneity hypothesis*[9]. Twinning would thus be an adaptive strategy more frequently expressed by females who are able to withstand the costs of twinning and reap the benefits[7,9].

The study of how twinning rates relate to fertility and survival not only shapes our understanding of why the frequency of twins varies within and between populations, but also guides medical research. For example, in the search for genetic variants underpinning female fertility, key studies[13,14] rely on the direct comparison of genomes from groups of twinners and non-twinners, because they are assumed to differ in their overall intrinsic fertility.

While the negative effects of twinning on maternal and offspring survival are indisputable, conclusions that twinning is positively associated with intrinsic fertility are questionable. These conclusions are based on numerous studies showing that when counting a twin birth as one birth, twinners have more births during their lifetime (*total births*) than non-twinners[5,8–10]. Crucially, however, inferring a positive correlation between intrinsic fertility and twinning propensity from a positive correlation between realisations of these latent traits—total birth and lifetime twinning status—constitutes an *ecological fallacy*[21–23].

An ecological fallacy[24] is a particular type of erroneous inference where patterns revealed in between-group comparisons are presumed to also apply to the individual data constituting such groups[25]. This well known and yet lasting problem caused by overlooking insidious effects of analysing sets of aggregated data plagues the conclusions from many studies across a range of fields[26,27].

Authors of previous work on the relationship between twinning and fertility (including most of us) have fallen into this aggregation trap by classifying each mother as a twinner or non-twinner based on an aggregation of all her birth outcomes prior to statistical analysis. A mother's lifetime twinning status confounds her twinning propensity with her exposure to the risk of twinning, which accumulates with the number of births she experiences[22]—just as people who drive as part of their job (e.g. taxi drivers) are more likely to be involved in a car crash because they drive more[28]. Not controlling for variation in exposure between mothers makes any comparison of twinners and non-twinners problematic. For example, consider the prediction that "twinning mothers should exhibit additional features of a robust phenotype, including shorter average inter-birth intervals, later ALB [age at last birth] and longer reproductive spans resulting in higher parities"[9]. Here, the predicted differences between twinners and non-twinners may arise in the absence of an association between intrinsic fertility and twinning propensity: women with shorter interbirth intervals, later age at last birth, or longer reproductive spans are more likely to have twins because they give birth more often.

Although the extent to which conclusions are robust to the effect of the aggregation will vary among studies, large biases are likely to be common[29,30]. Any study investigating which properties of an individual influence their risk of a given event (e.g. being involved in a traffic accident, catching a sexually transmitted disease, winning the lottery, or having twins) in a situation where the exposure to such risk varies, needs to account for variation in exposure (e.g. distance driven, number of sexual partners, number of tickets bought, or total births). Without this, it remains unknown whether twinners have, relative to non-twinners, a higher twinning propensity (which increases the risk of twinning at each birth), a higher intrinsic fertility (which increases the exposure to the risk of twinning), or both (as commonly assumed).

Assessing the nature of the relationship between twinning propensity and intrinsic fertility requires not only an analytical framework that accounts for differences in risk exposure within and between mothers, but also large amounts of reliable data. Modern datasets conflate natural variation in twinning propensity and intrinsic fertility with variation introduced by medical intervention and family planning, whereas historical datasets for single populations are often too small to study rare events such as twinning with precision (a notable exception being the dataset from the Utah population studied by Robson & Smith[9,23]). We thus assembled a unique demographic dataset by pooling together data from nine European populations (Table 1), resulting in 105,833 births to 21,290 mothers born in the 18th and 19th centuries[5,8,31–36]. These mothers were from populations that had not yet undergone much of the demographic transition towards lower lifetime number of offspring that is characteristic of industrialised populations[37]. For the sake of simplicity, we considered all births resulting in more than one child as twinning events. Such events therefore include rare cases of triplets. Furthermore, although our dataset does not distinguish dizygotic from monozygotic twins, estimations based on the sex of children suggest that the vast majority of twins in our dataset (~80%) were dizygotic, similar to other European populations[1,3].

We start this study by replicating previous analyses of the relationship between lifetime twinning status and total births after aggregating birth outcomes within mothers (i.e. without accounting for variation in risk exposure between mothers). Second, we examine the relationship between twinning and fertility at the level of births (i.e. using non-aggregated data) to

**Table 1 Details of data used in the present study, for each population separately and for all populations combined.**

| Population | Locations | Maternal birth period | Mothers | Non-twinners | Twinners | Twinner rate (‰) | Births | Singleton births | Twin births | Twinning rate (‰) | Offspring birth period | Total births (min-median-max) | References |
|---|---|---|---|---|---|---|---|---|---|---|---|---|---|
| Finland East | Jaakkima, Rautu | 1742-1899 | 871 | 781 | 90 | 103.33 | 4444 | 4347 | 97 | 21.83 | 1771-1940 | 1-7-17 | 33,34 |
| Finland Lapland | Inari, Enontekiö and Sodankylä | 1700-1884 | 757 | 695 | 62 | 81.9 | 3548 | 3482 | 66 | 18.6 | 1725-1918 | 1-6-13 | 8 |
| Finland SW-Archipelago | Hiittinen, Kustavi, Rymättylä | 1709-1899 | 2737 | 2443 | 294 | 107.42 | 12136 | 11816 | 320 | 26.37 | 1732-1942 | 1-6-15 | 3,6 |
| Finland West | Ikaalinen, Pulkkila, Tyrvää | 1700-1899 | 5669 | 5200 | 469 | 82.73 | 30733 | 30224 | 509 | 16.56 | 1721-1943 | 1-7-16 | 33,34 |
| Krummhörn | Lower Saxony, Germany | 1705-1823 | 3739 | 3461 | 278 | 74.35 | 17634 | 17336 | 298 | 16.9 | 1725-1868 | 1-6-17 | 5 |
| Sami Lapland | Inari, Enontekiö and Sodankylä | 1703-1880 | 957 | 885 | 72 | 75.24 | 4858 | 4780 | 78 | 16.06 | 1729-1920 | 1-7-13 | 8,32 |
| Sweden Lapland | Karesuando, Jukkasjärvi, Jokkmok, Vilhelmina and Jällivaara | 1721-1878 | 1943 | 1797 | 146 | 75.14 | 11106 | 10946 | 160 | 14.41 | 1749-1902 | 1-8-17 | 32,35,36 |
| Switzerland | Linthal, Elm | 1700-1899 | 4617 | 4394 | 223 | 48.3 | 21374 | 21138 | 236 | 11.04 | 1720-1945 | 1-6-18 | 31 |
| All the above | All the above | 1700-1899 | 21290 | 19656 | 1634 | 76.75 | 105833 | 104069 | 1764 | 16.67 | 1720-1945 | 1-7-18 | This paper |

obtain an estimate of the relationship no longer confounded by risk exposure. Third, we test four mechanisms that may drive the true (i.e. per-birth) relationship between twinning and fertility by simulating the reproductive histories of mothers under alternative biological scenarios, and comparing the simulated to the observed relationship. Finally, we turn to quantifying the effect of twinning propensity on the lifetime number of offspring produced—an integrative measure of lifetime reproductive success encompassing both total births and the increase in offspring number caused by multiple birth events.

## Results

**Relationship between twinning and total births.** In accordance with previous work[5,7–10], our data show that twinners had more total births than non-twinners (extra births to twinners: 1.43; $CI_{95\%}$: 1.22, 1.65; Fig. 1a; Supplementary Table 1), with the odds of being classified as a twinner increasing by 1.17-fold ($CI_{95\%}$: 1.16, 1.19; Fig. 1b; Supplementary Table 2) with each additional total birth. As 92.6% of twinners produced twins only once, the average number of extra births being larger than one shows that twinners produced, on average, more singletons than non-twinners.

The positive relationship between lifetime twinning status and total births may not, however, reflect a positive relationship between per-birth twinning probability and maternal total births[22]. Indeed, when we analyse the twinning status of each birth, we find that the per-birth twinning probability is negatively related to total births. That is, mothers who were more likely to have twins in any given birth event had fewer births overall (Fig. 2; Supplementary Table 3), with the odds of twinning being multiplied by 0.967 ($CI_{95\%}$: 0.952, 0.983) with each additional total birth. There is little evidence for variation in the slope of this relationship among populations, irrespective of whether such variation is modelled by means of an interaction between fixed effects (Likelihood Ratio Test, $X^2 = 11.9$, $p = 0.12$) or as a random slope (LRT, $X^2 = 1.22$, $p = 0.31$; see "Methods" for details).

**Candidate mechanisms shaping the relationship between twinning and total births.** The relationship depicted in Fig. 2 raises a new question: how does the negative relationship between per-birth twinning probability and total births come about? One possibility is that maternal age, which we explicitly did not control for in the previous analysis, drives the relationship due to its influence on the per-birth probability of twinning[1,11]. For example, mothers that start reproducing from the age of 30 onwards show an increased probability of twinning but will have a short reproductive lifespan and few total births. However, other mechanisms may also be at play, including some where maternal age influences both the per-birth probability of twinning and fertility component(s), and it is not a priori obvious how well different hypothesised mechanisms may explain the focal relationship.

Building on the knowledge of human life histories, and the effects of maternal age and parity on human twinning in particular[1,5,9,11,20,23,38,39], we hypothesised four non-mutually exclusive mechanisms that might shape the relationship between per-birth twinning probability and total births (Fig. 3). Each of these hypothesised mechanisms combine three generalised linear mixed-effects models (GLMM)—modelling, respectively, the three key life-history events recorded in our dataset: *parity progression* (whether or not a mother reproduces again after having given birth), *interbirth interval* (the duration between two consecutive births) and the *twinning outcome* of a birth (whether a birth yields a singleton or multiple children). All hypothesised

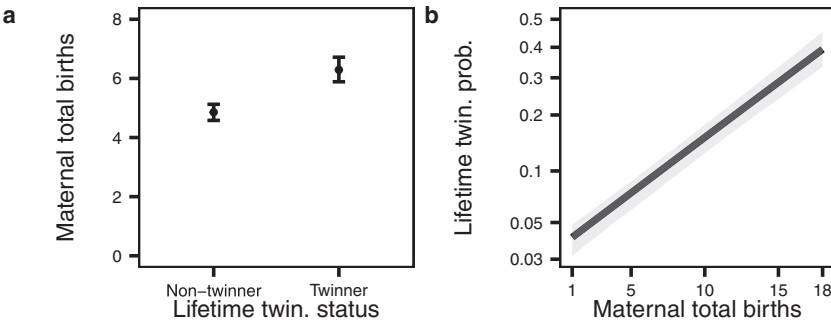

**Fig. 1 Relationship between twinning status and the number of maternal total births.** The plots represent marginal predictions (**a**: points, **b**: line) ± $CI_{95\%}$ (**a**: error bars, **b**: grey area) from the fits of generalised linear mixed-effects models including the variable depicted on the x-axis as fixed effect and variation between populations as a random effect ($n = 21,290$ mothers in total, from 8 populations; see Table 1). **a** Twinners had more total births (6.29; $CI_{95\%}$: 5.89, 6.72) than non-twinners (4.86; $CI_{95\%}$: 4.58, 5.12), and (**b**) the probability of a mother being a twinner was positively related to a mother's total number of births. Note that the y-axis is represented on a logit scale so as to display the outcome of the logistic (binary) regression as a straight line with an estimated slope β of 0.162 ($CI_{95\%}$: 0.145, 0.178). Model summary statistics are provided in Supplementary Tables 1, 2.

mechanisms account for the possibility of reproductive senescence via the consideration of the fixed effects of maternal age and parity (i.e. the number of past births) on parity progression and interbirth intervals, as well as the presence of maternal heterogeneity in intrinsic fertility by means of a "maternal identity" random effect. They differ in how twinning outcomes and twinning propensity relate to the two fertility components (parity progression and interbirth intervals). We first describe and illustrate each mechanism before proceeding to formally testing them using individual-based simulations.

In mechanism P, a twinning event impacts Parity progression. A drop in parity progression after a twinning event is expected since such an event may increase the risk of maternal death and reduce reproductive health[3,5,40]. Furthermore, mothers may show a reduced inclination to continue reproducing after the birth of twins, as suggested for some modern populations[41, but see 42]. Fitting a GLMM to our parity progression data (see "Methods") confirmed that mothers were indeed less likely to keep reproducing following the delivery of twins versus a singleton (Fig. 4a; Supplementary Table 4). The estimated effect of a twinning event on parity progression was independent of the negative effect of maternal age and parity number, which were included as covariates in this statistical model so as to capture ovarian ageing[43]. Mechanism P is thus consistent with a negative relationship between per-birth twinning probability and total births.

In mechanism I, a twinning event impacts Interbirth intervals. On the one hand, raising twins to maturity requires more maternal investment (e.g. the production of around twice as much milk during the first 9 months of lactation[44]), which may extend the duration of lactational amenorrhoea and delay the next reproduction. On the other hand, the higher mortality of twin offspring[1,5,7,16–18,45] may lead to shorter interbirth intervals. Fitting a GLMM to the duration of interbirth intervals reveals that they tended to be slightly shorter after a twinning event (shorter by 1.03 months at mean age and mean parity; $CI_{95\%}$: −1.98, −0.194; Fig. 4b; Supplementary Table 5). As with mechanism P, this effect of a twin event was independent from the effect of maternal age and parity number. Since shorter interbirth intervals after giving birth to twins allow for more total births, the effect of mechanism I goes against the effect of mechanism P in shaping the relationship between per-birth twinning probability and total births.

Mechanism S captures the effect that the reproductive Schedule of a mother exerts on her probability of having twins. In particular, it considers that maternal age influences both the per-

birth twinning probability (S1) and total births (S2), giving rise to a relationship between these variables. This could happen, for example, because women starting to reproduce late in their life are more likely to both produce twins and have fewer births (as mentioned above). As for the other life-history traits, we quantified the effect of maternal age on the per-birth twinning probability while controlling for parity since both variables are strongly correlated (Spearman's rho = 0.69 in our dataset) and potentially associated with twinning[1]. The GLMM fitted to our data is consistent with S1: we replicated the well-established result that the maximal per-birth twinning probability is reached when mothers were in their mid to late thirties[11] (Fig. 4c; Supplementary Table 6). The model fit also reveals that the highest per-birth twinning probability was reached at first birth, with little difference across other parities. This result is consistent with previous studies[46], although others have also reported no effect[5,7], or a positive effect of parity on twinning probability[1,47]. The origin of such variation remains mysterious because studies vary in how they attempt to disentangle the effect of the two correlated variables and this methodological variation obfuscates possible biological differences across populations. Data also support S2: both components of total births (parity progression and interbirth intervals) were associated with maternal age and/or parity. Specifically, the probability of parity progression decreased with maternal age and parity (Fig. 4a), while the duration of interbirth intervals mainly increased with parity (Fig. 4b). The observation that twinners started to reproduce later than non-twinners is supportive of mechanism S, even though this reproductive delay only applied to twinners who totalled one birth in their life (Supplementary Fig. 1; Supplementary Table 7). Altogether, these results suggest that mechanism S could also have contributed to the observed relationship between per-birth twinning probability and total births.

Mechanism H focuses on the role of maternal Heterogeneity and posits that mothers who have a higher twinning propensity (i.e. an overall higher per-birth twinning probability irrespective of age or parity) are also more fertile overall (i.e. higher intrinsic fertility), either through enhanced parity progression, shorter interbirth intervals or both. Supportive of the presence of maternal heterogeneity, the fits of the three GLMMs modelling each life-history event show random effect variance between mothers (see Supplementary Tables 4–6). In contrast to what is assumed by the heterogeneity hypothesis, however, the predictions for these random effects show that mothers with high random effects on twinning propensity experienced low random effects on parity progression, and high random effects on

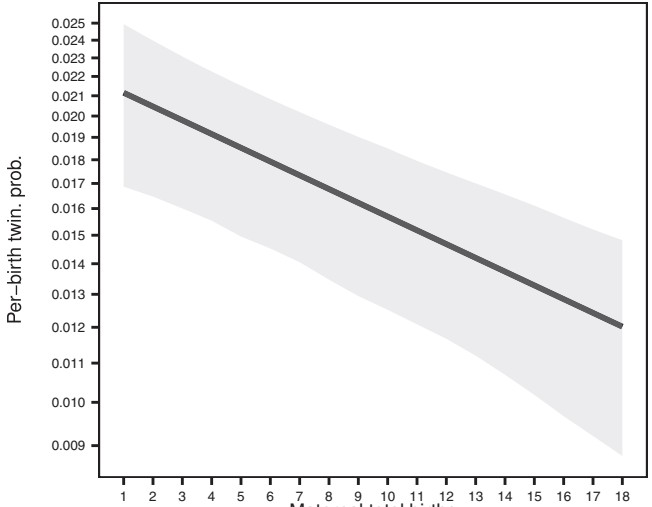

**Fig. 2 Relationship between per-birth twinning probability and maternal total births.** This plot shows marginal predictions (line) ± CI$_{95\%}$ (grey area) from the fit of a generalised linear mixed-effects model including maternal total births as fixed effect and variation between populations as a random effect ($n = 21,290$ mothers in total, from 8 populations; see Table 1). Estimates obtained for per-birth twinning probability vary between 0.021 (CI$_{95\%}$: 0.017, 0.025) for mothers who gave birth only once, and 0.012 (CI$_{95\%}$: 0.0088, 0.015) for mothers who gave birth 18 times—the maximal total births recorded in our data. Note that the y-axis is represented on a logit scale so as to display the outcome of the logistic (binomial) regression as a straight line with an estimated slope β of −0.0338 (CI$_{95\%}$: −0.0510, −0.0168). Model summary statistics are provided in Supplementary Table 3.

interbirth intervals (Supplementary Fig. 2). Maternal heterogeneity thus exerted a negative influence on the relationship between per-birth twinning probability and total births.

In sum, all four mechanisms potentially influenced the relationship between per-birth twinning probability and total births: the impact of twinning events on parity progression (mechanism P) and the role of heterogeneity between mothers (mechanism H) are likely to have contributed to its negative slope, while the impact of twinning events on interbirth intervals (mechanism I) is likely to have pushed the relationship in the other direction. The effect of the schedule of reproduction (mechanism S) is more ambiguous as it could have influenced the relationship either way depending on the exact reproductive schedule of mothers.

**Contribution of the candidate mechanisms to the relationship between twinning and total births.** To assess the role of each of these four mechanisms in shaping the relationship between per-birth twinning probability and total births, we used the estimates provided by the three fitted GLMMs illustrated in Fig. 4 (as well as sub-models derived from them; see Methods for details, as well as Supplementary Tables 8–12) to parametrise individual-based simulations. We programmed the simulations to replay in silico the reproductive history of the women from our dataset, birth after birth, by drawing random realisations of the three key life-history events in the presence or absence of each of the four mechanisms (Supplementary Figs. 3, 4). After simulating multiple datasets under each of the 16 possible combinations of the four mechanisms (PISH, PIS, PIH, PSH, PI, …, Fig. 5), including a scenario that does not contain any such mechanism (named 0), we quantified the relationship between per-birth twinning probability and total births in each simulated dataset, and

compared these to the relationship observed in the empirical data. We performed this comparison using a so-called goodness-of-fit test which considered, in turn, each simulation scenario as a null hypothesis which the empirical data had a chance to reject (see Methods and Supplementary Notes for details). This approach is suitable for identifying those hypothesised scenarios that can explain the data, and for excluding those that cannot. Importantly, accepting any single scenario from the set of those retained does not imply that it is the true one. Furthermore, none of the scenarios will be retained if all fail to fit the data adequately.

Our analysis of the simulated datasets revealed that the relationship shown in Fig. 2 is most likely the result of the impact of a twinning event on parity progression (mechanism P) and to a lesser degree on interbirth intervals (mechanism I), as well as the effect of the schedule of reproduction on per-birth twinning probability (mechanism S). Indeed, all six simulation scenarios not rejected by our goodness-of-fit test (i.e. $p > 0.05$) include mechanism P (scenarios P, PI, PS, PIS, PSH, PISH; Fig. 5; Supplementary Table 13) and four of them include mechanism S (PS, PIS, PSH, PISH). Simulating heterogeneity in twinning propensity between mothers (mechanism H) did not increase the goodness of fit of the eight simulation scenarios that did not initially consider this mechanism (0, P, I, S, PI, IS, PIS), with the exception of the goodness of fit for PS, which was marginally improved. Moreover, the scenario with the best goodness of fit includes mechanisms P, I and S (PIS; $p = 0.169$) but not H, which shows that the negative association we measured between twinning propensity and intrinsic fertility contributed little to the negative relationship between per-birth twinning probability and total births. Because mechanism I contributes positively to the relationship (Figs. 4b, 5), the analysis reveals that P and S are the only mechanisms, out of the four considered, that could be responsible for the negative association between per-birth twinning probability and total births.

**Twinning and total number of offspring.** While our results show that an increase in per-birth twinning probability led to a reduction in total births, the overall effect of twinning on the reproductive output of women may still be positive when counting the total number of offspring produced. This is because, by definition, twinning events lead to additional offspring per birth (one in the case of twins *sensu stricto*, two in the case of triplets). To test whether twinning propensity increases the total number of offspring irrespective of the variation in risk exposure between mothers, we need to vary this trait while keeping all other factors the same. We therefore simulated the reproductive histories of mothers under scenario PIS (the one best fitting the data) after increasing the per-birth probability of twinning across all maternal ages and parity values while keeping other effects acting on fertility unchanged, and compared the results of such simulations to those produced under the original twinning probabilities.

The original twinning probabilities lead to the simulation of mothers with a mean (95% Central Range from simulation replicates, CR$_{95\%}$) twinning rate of 16.7 (15.8, 17.6)‰, a twinner rate (i.e. the frequency of mothers that are twinners) of 77.1 (73.7, 81.6)‰, a mean number of births of 4.83 (4.81, 4.86), and a mean number of offspring of 4.91 (4.88, 4.95). Mothers simulated to have an approximately 10-fold higher twinning rate (mean twinning rate: 167‰, CR$_{95\%}$: 165, 169; mean twinner rate: 535‰, CR$_{95\%}$: 530, 541) would have more offspring (mean = 5.54; CR$_{95\%}$: 5.50, 5.58) despite fewer total births (mean = 4.75; CR$_{95\%}$: 4.72, 4.78). The magnitude of the simulated increase in twinning rate is arbitrary: an increase in twinning propensity

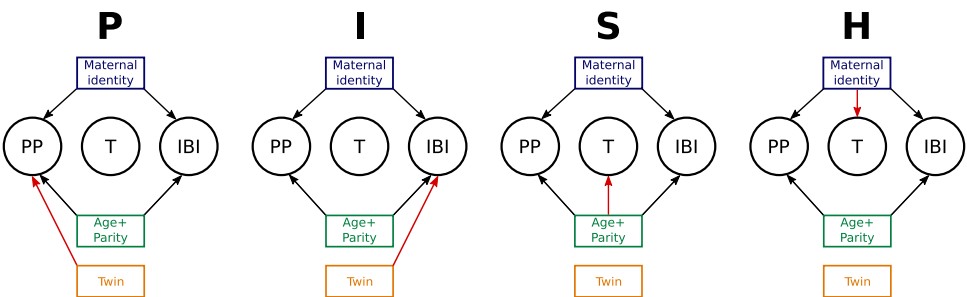

**Fig. 3 Representation of the four, non-mutually exclusive, candidate mechanisms impacting the relationship between per-birth twinning probability and total births.** Circles represent the three life-history events we considered: parity progression (PP), per-birth probability of twinning (T) and interbirth interval (IBI). The rectangles represent the variables potentially shaping these life-history events—maternal age and parity at a given birth (referred to as Age + Parity) and whether the last birth was a twin birth or not (Twin)—as well as a random effect capturing other sources of heterogeneity between mothers (Maternal identity). Black arrows represent relationships assumed in all simulation scenarios. Another random effect capturing differences between populations was also considered for all life-history events and all mechanisms (not shown). Red arrows represent relationships used to activate each mechanism.

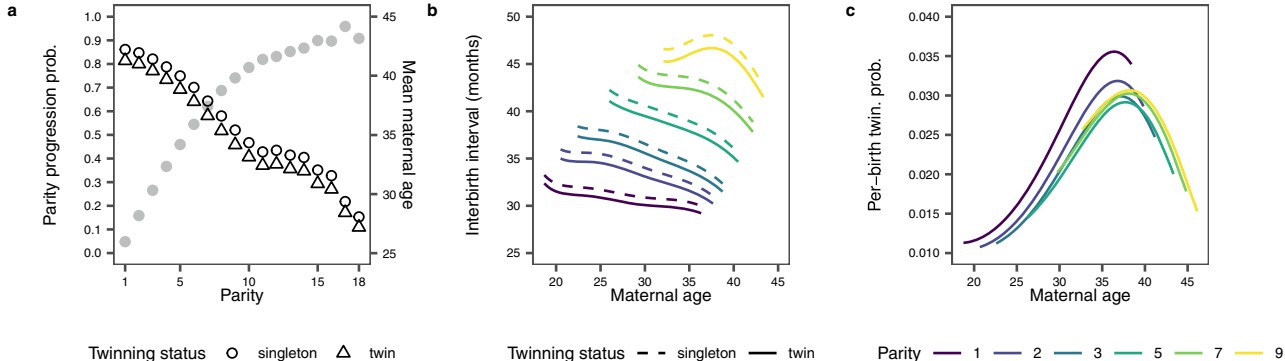

**Fig. 4 Relationship between maternal age, parity and twinning status and the three life-history events describing maternal reproductive life. a** The probability that a mother keeps reproducing after a given parity (parity progression) was lower following the birth of twins than following the birth of a singleton ($n = 105{,}833$ births in total, from 21,290 mothers). Marginal predictions for the probability of parity progression are shown in black with open symbols, with scale on the left $y$-axis, and were computed given a parity and twinning status as average predictions over the empirical distribution of maternal age at each parity. The mean of maternal age for each parity is shown by the grey line with filled symbols, with the scale on the right $y$-axis. **b** The duration between two consecutive births (interbirth interval) increased with parity (colour) but slightly decreased after the birth of twins vs singleton (line type) or with maternal age ($x$-axis) when parity is held constant ($n = 84{,}543$ births in total, from the 18,520 mothers who had at least two births). **c** Mothers presented the highest risk of twinning ($n = 105{,}833$ births in total, from 21,290 mothers) during their mid and late thirties, as well as at parity 1 (purple line). For (**b**, **c**), marginal predictions are shown as curves and are computed for maternal age and parity values spanning the 95% central ranges of the observed distributions of these variables. Model summary statistics are provided in Supplementary Tables 4–6.

always increases the number of offspring, but a large increase exacerbates the differences.

Although the total number of offspring born increased with twinning propensity, a trade-off exists between the number of offspring from a given birth and the survival prospects of such offspring. This is true in humans[1,5,7,16–18,45], as well as in non-human primates[48] and many other species[49,50]. Although the exact mortality costs of twinning vary with time and space[2,18], as well as the cut-off used for the age at which mortality is compared[1], if twins are less likely to survive birth, infancy and/or childhood than singletons, an increase in twinning rate may no longer be systematically associated with an increase in lifetime reproductive success.

When simulating the relatively small mortality difference observed between singletons and twin offspring as observed for some of the populations we sampled[8], the increase in twinning rate remained associated with an increase in total number of (surviving) offspring (baseline: 3.98, $CR_{95\%}$: 3.96, 4.00; 10-fold increase in twinning rate: 4.23, $CR_{95\%}$: 4.20, 4.26). In contrast, when we used larger estimates for the mortality difference between singletons and twin offspring as documented for other populations we sampled[3],

the increase in twinning rate became associated with a decrease in total number of surviving offspring (baseline: 3.41, $CR_{95\%}$: 3.39, 3.43; 10-fold increase in twinning rate: 3.33, $CR_{95\%}$: 3.31, 3.35). Therefore, when measured as the number of offspring reaching adulthood, whether an increased twinning propensity was associated with a higher or lower lifetime reproductive success depended on the mortality levels in the population considered. As above, this outcome holds irrespective of the exact increase in twinning propensity simulated.

## Discussion

In this paper, we investigated the relationship between twinning and fertility by analysing the reproductive history of >20k mothers from pre-industrial European populations. Altogether, we show that in pre-industrial Europe, twinning impacted fertility, and that these traits are not, as others have proposed, both indicators of the same underlying physiology.

Previous studies of the relationship between twinning and fertility compared women according to their lifetime twinning status (i.e. twinner vs non-twinner). When we applied this

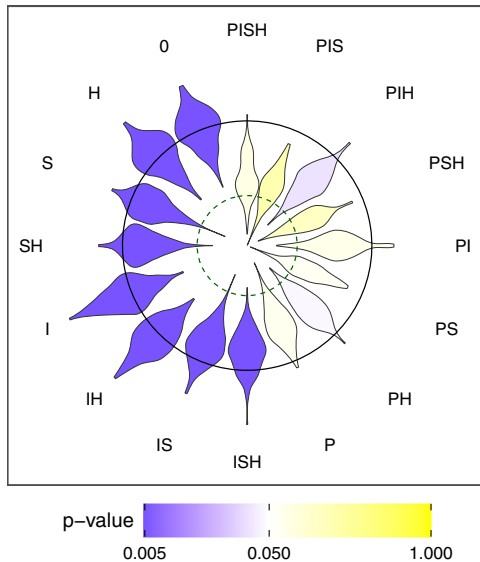

**Fig. 5 Goodness-of-fit for the 16 simulation scenarios testing which of the four mechanisms could account for the slope between the logit of the per-birth twinning probability and maternal total births shown in Fig. 2.** The letters P, I, S & H are used to indicate which mechanisms were included in each simulation scenario. Mechanism P considers that a twinning event impacted parity progression. Mechanism I considers that a twinning event impacts the interval between that birth and the next one. Mechanism S considers that the reproductive schedule of a mother impacts both her twinning probability and her total number of births, creating an association between these two variables. Mechanism H considers that twinning propensity and intrinsic fertility are associated as a consequence of maternal heterogeneity. The scenario 0 does not include any of these mechanisms. The shape of each petal of the flower plot shows the distribution of slopes obtained for simulation replicates under each scenario ($n = 200$ per scenario). The colour of each petal provides the result of a goodness-of-fit test. Yellow petals correspond to simulation scenarios compatible with the observed slope ($p$-value > 0.05) and purple petals correspond to scenarios incompatible with them ($p$-value ≤ 0.05), with variation in the intensity of the colour reflecting the $p$-value as shown in the colour bar. The dashed green inner circular contour line represents the slope shown in Fig. 2. A simulation scenario that would consider the true mechanisms would therefore show a yellow petal whose centre was intersected by this circle. The black outer circular contour line corresponds to a null slope with negative slopes falling within the circle and positive slopes being outside.

statistical approach, which by aggregating the outcomes of all births over a woman's life is biased by variation in the exposure to the risk of twinning between mothers, we observed the positive relationship between twinning and fertility that many researchers have interpreted as the consequence of a (positive) relationship between twinning propensity and intrinsic fertility[5,7-10]. In contrast, when we controlled for variation in risk exposure by performing the same analysis without aggregating data (i.e. at the level of each birth) to reveal the unbiased biological relationship between twinning and fertility, its sign flipped from positive to negative—an extreme effect of data aggregation known as "Simpson's paradox"[51,52].

The negative relationship between twinning propensity and intrinsic fertility was also found when correlating the estimates for the "maternal identity" random effect obtained by fitting hierarchical (mixed) models. However, a goodness-of-fit analysis showed that this relationship between latent traits exerted at best a minor influence on the relationship between realised twinning and fertility outcomes. This goodness-of-fit analysis suggests that

the latter relationship was not a consequence of maternal heterogeneity in common physiological factors underpinning both fertility and twinning, but rather a consequence of two mechanisms: i) mothers reproducing later than average were both more likely to produce twins and to have fewer births, and ii) mothers were more likely to cease reproduction after a twinning event, irrespective of the underlying cause.

Our results cast doubt on the validity of findings from epidemiological and clinical studies that assumed that the lifetime (dizygotic) twinning status is a proxy for female intrinsic fertility. For example, a number of studies[13,14,53–57] aimed at identifying gene variants associated with twinning and/or fertility. This comparison of the characteristics of women according to whether they have produced twins or not is misguided because it conflates twinning propensity with total births. When searching for genetic variants associated with female intrinsic fertility, this approach increases the risk of both false positives and false negatives. For example, alleles that appear to be associated with twinning may, in fact, be alleles functionally linked to high parity progression only, and not to twinning propensity. Similarly, alleles that increase (or decrease) twinning propensity may not be detected if they simultaneously decrease (or increase) intrinsic fertility. The problem is not restricted to genetic studies, however. For example, studies that have used (dizygotic) twinning prevalence to identify secular trends in intrinsic fertility have also failed to control for variation in risk exposure between women[58–60].

Our results also cast doubt on the validity of the heterogeneity hypothesis, which posits that females of high intrinsic fertility have a higher twinning propensity[9,23]. Importantly, the rejection of this hypothesis by our data does not imply the absence of heterogeneity among mothers: the three life-history traits we modelled (parity progression, interbirth interval and twinning) were all found to vary among mothers independently of their age and reproductive history. However, mothers with a high twinning propensity were, in general, not the same individuals as the mothers with a high intrinsic fertility. Although individuals with high intrinsic fertility and twinning propensity (i.e. "supermums"[7] or "robust maternal phenotypes"[9,61]) did exist, they were rare and not representative of the relationship between twinning and total births, at least in pre-industrial Europe. Whether the evidence documented by Robson and Smith[9,23] in support of the heterogeneity hypothesis for another pre-industrial population of European origin (Utah, USA) hold up to a reanalysis on non-aggregated data is as of yet unknown.

Correlations between lifetime twinning status and maternal characteristics other than total births have also been interpreted as further support for the heterogeneity hypothesis[7,9,23,38]. Again, however, this interpretation fails to distinguish between these characteristics influencing twinning propensity per se, and the exposure to the risk of twinning. For example, Sear et al.[7] reported that Gambian girls who become twinners have a higher body mass index (BMI) than girls who become non-twinners. Although this could be because individuals with high BMI have a higher twinning propensity, alternatively there may be a physiological association between fertility and BMI whereby a higher BMI would increase the exposure to the risk of twinning. This would result in a positive association between BMI and lifetime twinning status, even if BMI had a null (or even negative, as in the case of a Simpson's paradox) physiological association with twinning propensity.

If there is little evidence that within-population variation between mothers generates a relationship between twinning and fertility, the heterogeneity hypothesis does not provide a satisfactory explanation for why twinning rates are, despite the costs of twinning, not null, and why they show so much variation in space and time. Instead, our results are compatible with explanations which consider the role of variation within individual

mothers (i.e. across their reproductive lifetimes), as well as between populations. In particular, we observed a clear peak in twinning probability for women in their mid to late thirties. This well-documented pattern[1,5,11,38,39] is (qualitatively) predicted by the *ova insurance hypothesis*[4,11]. This hypothesis states that dizygotic twinning occurs as a by-product of polyovulation, a condition-dependent compensatory mechanism against embryo mortality that would thus be selected to increase with maternal age. It predicts women reproducing early in their life will tend to have more singletons because polyovulation is rare, and women reproducing late will tend to have more singletons because their polyovulation is masked by the high rate of embryo mortality[11]. The rate of egg production exceeding the rate of births, the rate of dizygotic twins exceeding the rate of monozygotic twins, and the frequent reabsorption of one fertilised egg after a successful double fertilisation (i.e. the "vanishing twin syndrome"[12]) all provide further indirect evidence in support of this hypothesis[4].

Our numerical simulations are compatible with between-population variation being another driver of the relationship between twinning and fertility: an increase in twinning propensity increased women's lifetime reproductive success when twin mortality was not much larger than that of singletons, but decreased it otherwise. Hence, the twinning rate that maximises women's lifetime reproductive success will depend on the population and its environment, and populations may thus evolve different twinning rates—an idea known as the *eco-evolutionary hypothesis*[6,20]. Testing this hypothesis, which requires twinning, mortality and fertility data sampled from populations located in widely different environments, is beyond the scope of this study, and attempts by others[5–8,10] relied on the problematic definition of twinning status by aggregating over an individual's life. Yet, our data show that the prerequisites for this hypothesis are met. Until the ova insurance hypothesis is shown to fully account for variation in twinning rates within and between populations, the eco-evolutionary hypotheses should not be disregarded.

Both the ova insurance hypothesis and the eco-evolutionary hypothesis are compatible with our finding that the relationship between twinning and fertility is not particularly strong once non-aggregated data are analysed. Indeed, under the ova insurance hypothesis one expects a weak relationship because selection acts upon double ovulation and not upon twinning per se. Under the eco-evolutionary hypothesis, a weak relationship is expected if populations are close to their evolutionary equilibrium. In contrast, the heterogeneity hypothesis would result in a stronger relationship established by condition dependence.

Irrespective of the hypothesis, trait, population and species under study, it is likely that much research in life science and medicine is impacted by the problem we have highlighted—an ecological fallacy caused by aggregating life-history events within individuals. Flagging which particular results or claims are robust to the effect of data aggregation requires reanalysis using statistical approaches that do not rely on aggregation, such as hierarchical models[27]. The most effective way to avoid the issue in the first place is for data providers to distribute non-aggregated datasets. The costs of more stringent ethical and legal requirements (whenever applicable, to respect individual privacy) and solving technical difficulties (such datasets are much larger than their aggregated counterparts) will pale in comparison to the benefits of avoiding ecological fallacies.

## Methods
### Data preparation
*Historical data.* The primary source of data is historical parish registers, which have been transcribed under the supervision of many of the study authors over a number of decades, primarily for evolutionary demographic research. Our dataset (Supplementary Data 1) includes nine European populations, including some for

which the positive relationship between maternal lifetime twinning status and maternal total births has been described previously[5,6,8,10]. Details for the populations used in this study are given in Table 1 and in Supplementary Table 14. The sourcing of each dataset and the socio-ecological background of each population have already been described in previous studies (see Table 1 for references). Overall, there is no reason to suspect a high level of consanguinity in these populations[62], so our analyses do not account for the variable level of relatedness between individuals. The datasets cover pre-industrial periods in which the lifetime reproductive success was high and the majority of people were living and working in agrarian communities, except for the Samis (from northern Finland and Sweden) who made their living fully or in part from a combination of herding, fishing and hunting. The smooth decline of the probability of parity progression with parity (Fig. 4a) suggests that mothers did not effectively limit their reproductive success with the aim of achieving a small family size, as found in populations that have undergone a demographic transition.

*Data selection.* We use the term *family* to describe a mother and all individuals to whom she gave birth over her life. For our analyses, all families considered met the following criteria: the mother's age was known at a monthly resolution and her life course traced until at least age 45 (approximating full reproductive life), the birth year and month of all offspring must have been recorded and consecutive births were all at least nine months apart from one another. In the case of one population (Norway) and of a few observations in the other populations, the month of birth was not available. These data were thus not considered in the results presented in main text because some of our analyses require an accurate estimation of the interbirth interval. Most analyses are thus based on data from eight populations. Nevertheless, the slope of the negative relationship between twinning and total birth remained very similar irrespectively of whether or not such data were included (Supplementary Fig. 5), which suggests that the exclusion of Norwegian data does not alter our main conclusions. Information on the populations considering also the data for which the birth months were missing is provided in Supplementary Table 14.

*Twin identification.* In our data, the maximum number of offspring to constitute a multiple birth was three. We use the term *twin(s)* to refer to offspring who were the result of the same multiple birth (including 1745 sets of twins *sensu stricto* and 19 sets of triplets in the filtered dataset and, respectively, 1915 and 20 in the dataset including observations that lack birth months). Although twins are sometimes explicitly indicated in the data sources, this is not always the case. Thus, for the sake of consistency across our populations, twin births were identified when at least two individuals born to the same mother appeared with similar birth dates, according to strict criteria: if the exact birth dates were available, then offspring were identified as twins if their birth dates were no more than one day apart. If the exact birth dates were not available, then an identical birth year and month were considered sufficient for positive twin identification.

### Analyses and simulations
*Characterisation of the relationship between twinning and fertility.* We began by characterising the relationship between lifetime twinning status and maternal total births by fitting two models. For the first, we used a Generalised Linear Mixed-effects Model (GLMM) to investigate whether the mothers of twins (twinners) had experienced a larger or smaller number of births than mothers who only had singletons (non-twinners). We fitted this GLMM on the mother-level data with the R package spaMM[63] using the call:

$$\text{fitme}\big(\text{births\_total} \sim 1 + \text{twinner} + (1|\text{pop}),$$
$$\text{data} = \text{mother\_level\_data}, \tag{1}$$
$$\text{family} = \text{Tnegbin}(\text{link} = \text{"log"})\big)$$

The response variable births_total refers to all births recorded over a mother's observed lifetime (count data). The term 1 informs the function to fit an intercept (which happens by default, but is indicated here for clarity). The predictor twinner refers to maternal lifetime twinning status (binary: twinner vs non-twinner) and is modelled as a fixed effect. The term pop refers to the population identity (qualitative variable with eight levels) and is modelled by a Gaussian random effect acting on the intercept, which allows for the modelling of the heterogeneity between populations that is not captured by the fixed effects. The argument family is used to define the error structure and the link function of the GLMM (more on this below).

In a second model, we reversed which variable is used as a response and which is used as the fixed-effect predictor. This allowed us to analyse how maternal total births predicted the probability of a mother producing twins during her lifetime using the call:

$$\text{fitme}\big(\text{twinner} \sim 1 + \text{births\_total} + (1|\text{pop}),$$
$$\text{data} = \text{mother\_level\_data}, \tag{2}$$
$$\text{family} = \text{binomial}(\text{link} = \text{"logit"})\big)$$

The fitted models 1 and 2 are depicted in Fig. 1a, b and Supplementary Tables 1, 2. While models 1 and 2 represent two sides of the same coin, the fit of both models is justified because each model formulation provides complementary information:

expressing the effect of twinning on fertility relates to previous work[5–10,57] and expressing the effect of fertility on twinning is a first step toward identifying what shapes twinning propensity, the focus of this paper.

For the model predicting total births (model 1), we chose to use a negative binomial error structure. Using this error structure produced a fit of the data that was better than a (truncated) Poisson—the usual alternative for count data—as evidenced by much smaller marginal and conditional AIC values[64]. Here we specifically used a *truncated* negative binomial distribution because the data do not possess zeros by construction (only mothers are present in the dataset, i.e. there are no nulliparous women). For the model predicting lifetime twinning status (model 2), we chose a binomial error structure which is appropriate for binary data.

Modelling the proportion of twin births among all births per mother is an effective way to avoid biases caused by differences in exposure to the risk of having twins affecting the relationship between twinning and fertility. For this, we fitted the following third model:

$$\text{fitme}\big(\text{cbind (twin\_total, singleton\_total)}$$
$$\sim 1 + \text{births\_total} + (1|\text{pop}),$$
$$\text{data} = \text{mother\_level\_data},$$
$$\text{family} = \text{binomial (link} = \text{``logit''})\big) \tag{3}$$

In this model, the variable twin_total refers to the mother's total number of twin births (i.e. one for each twinning event), singleton_total refers to the lifetime number of singleton births, and the cbind() function serves to indicate the fitting function to model the frequency of twinning events based on these two variables, which is interpreted as number of successes and failures of a binomial experience. The fitted model 3 is depicted in Fig. 2 and Supplementary Table 3.

We modified model 3 so as to test whether the effect of total births differed significantly between populations. To do so, we considered that the effect of populations on total births could either be modelled as an interaction between fixed effects or as a random slope. For the former representation, we thus compared the fit of a model with linear predictor structure defined in spaMM as 1+births_total*pop to that of a model with the structure 1+births_total+pop. For the latter representation, we compared the fit with linear predictor structure 1+births_total + (1|pop) (i.e. model 3 as introduced above) to that of 1+births_total + (1+births_total|pop). We performed this testing procedure by comparing the likelihood ratio between each pair of alternative fits to the expectation of such a ratio under the null hypothesis. The distribution of the statistics used for the test was computed using 1000 parametric bootstrap replicates, which we generated using the function anova() provided by spaMM[63]. The test revealed a small non-significant variation in slopes between populations (see Results). For the sake of simplicity, we thus considered the effect of births_total the same across populations in all other analyses.

*Modelling life-history events using GLMMs.* To reveal the biological mechanisms responsible for the relationship between twinning and fertility, we first fitted statistical models describing how age, parity and twin/singleton status, as well as individual and population differences influenced three key life-history events: parity progression (PP), the duration of interbirth intervals (IBI) and the twinning outcome of births (T). These models were fitted on birth-level data by the following calls:

$$\text{fitme}\big(\text{PP} \sim 1 + \text{twin} + \text{poly(cbind(age, parity), best\_order)}$$
$$+ (1|\text{maternal\_id}) + (1|\text{pop}),$$
$$\text{data} = \text{birth\_level\_data},$$
$$\text{family} = \text{binomial (link} = \text{``logit''}) \tag{4}$$

$$\text{fitme}\big(\text{IBI} \sim 1 + \text{twin} + \text{poly (cbind(age, parity), best\_order)}$$
$$+ (1|\text{maternal\_id}) + (1|\text{pop}),$$
$$\text{data} = \text{birth\_level\_data},$$
$$\text{family} = \text{negbin (link} = \text{``log''})\big) \tag{5}$$

$$\text{fitme}\big(\text{T} \sim 1 + \text{poly(cbind(age, parity), best\_order)}$$
$$+ (1|\text{maternal\_id}) + (1|\text{pop}),$$
$$\text{data} = \text{birth\_level\_data},$$
$$\text{family} = \text{binomial (link} = \text{``logit''})\big). \tag{6}$$

The response variables of models 4, 5 and 6 are thus PP, IBI and T, which refer to whether the mother went on to reproduce again or not (a boolean), the duration of the interbirth interval between the focal birth and the next (a discrete number of months) and whether the birth resulted in twins or not (a boolean), respectively. In addition to the terms that have already been defined, we now have the term poly(cbind(age, parity), best_order) to code for a polynomial describing the effect of maternal age, parity and their possible interaction. The two-variable polynomial function was applied on maternal age (with a monthly resolution) and parity (i.e. the current birth rank). Such a polynomial term allowed us to explore the influences of maternal age and parity on each response variable while encompassing the non-linearity of these predictors. We also have the predictor variable twin, which is a boolean that indicates if the previous birth event of a given

mother resulted in twins or not (the variable twin and T are the same, but we used two different names to clarify when it is used as a response or as a predictor). Finally, we have the random effect "maternal identity" (maternal_id), which is used to represent intrinsic variation among mothers, that is, heterogeneity of expected response among individuals, beyond that due to the fixed effects and the population random effect. This random effect therefore measures maternal intrinsic fertility (in models 4 & 5) and twinning propensity (in model 6).

To determine the best polynomial order (best_order) for the polynomial term we attempted orders from 0 to 6 and selected, for each model, the order leading to the model fit associated with the smallest marginal AIC. A polynomial of order 6 is sufficient to fit very complex shapes. Polynomial orders obtained by this procedure are given in the summary tables of the model fits given in Supplementary Tables. Importantly, maternal age and parity are highly correlated together (Spearman's rho = 0.69), unequally correlated to response variables and exert non-linear effects. These are precisely the conditions in which collinearity issues are the most severe[65]. This justifies why we considered them jointly in all statistical models, as well as why we did not attempt to partition their respective biological effects in our analyses (except for the visual representation in Fig. 4).

In order to study how the lifetime twinning status influenced maternal age at first birth, we also fitted the following model:

$$\text{fitme}\big(\text{AFB} \sim 1 + \text{twinner} * \text{births\_total\_fac} + (1|\text{pop}),$$
$$\text{data} = \text{mother\_level\_data},$$
$$\text{family} = \text{negbin(link} = \text{``log''})\big) \tag{7}$$

In this model, the response variable AFB corresponds to the age at first birth expressed as a number of months (discrete data) and the predictor births_total_fac corresponds to a qualitative variable referring to maternal total births (10 levels: 1, 2, …, 9, 10 + ). We here considered a possible interaction between twinner and births_total_fac. We used the negative binomial family as in model 1 but as for model 5 there is no need to consider here the truncated form of the distribution. All other terms have already been defined. The fitted model is depicted in Supplementary Fig. 1 and Supplementary Table 7.

*Marginal predictions for GLMMs.* All predictions shown in plots or given in text represent *marginal predictions*. This means that the predictions for the quantities of interest (maternal lifetime births, twinning probabilities and age at first birth) are a function of coefficients of the fixed effects, and of the variance of the random effects. To be more precise, we averaged, over the fitted distribution of random effects, the predictions expressed on the scale of the response (i.e. back-transformed from the scale of the linear predictor) and conditional on the fixed and random effects. Unlike the traditional *conditional predictions* computed for a specific value of the random effects (often 0), such computation provides unbiased predictions and should be favoured in the context of GLMMs where random effects act non-additively on the expected response (which is the case when the link function of the model is not identity[66]. We estimated 95% intervals for these marginal predictions (CI$_{95\%}$) using parametric bootstraps with the help of the function spaMM_boot() from the R package spaMM and boot.ci() from the R package boot. More details can be found by looking at the code of the functions compute_predictions() and compare_prediction() in our supporting R package twinR (see Code availability).

*Simulating the life history of mothers.* We produced an individual-based simulation model of human female life history to investigate the contribution of four mechanisms to the relationship, shown in Fig. 2, between per-birth twinning probability and maternal total births—an approach generally known as pattern-oriented modelling[67]. Each simulation proceeds in the following way: first, we initialised the simulation with representations of the exact same mothers present in the observed dataset, setting their population and maternal identities as the real ones, their starting ages at the observed values for age at first birth and their parity to one. Following this initialisation, the virtual lives of mothers proceeded as multiple iterations of a sequence of three life-history events, informed by statistical models (see below) and subject to the hypothesis being tested (Supplementary Figs. 3, 4). Specifically, for each mother, the twin/singleton status (T) of the current birth was first determined using a GLMM predicting T. Then, whether or not she will go on to reproduce at least once more was determined by simulating her parity progression status (PP) using a GLMM predicting the parity progression probability. For mothers who do continue reproducing, we finally used a third GLMM to determine the length of the interval between the current birth and the next one (IBI). At each iteration, a mother's parity is increased by one, and age is increased by the simulated length of the interbirth interval. All predictions were performed conditionally on the value for the predictor characterised by both fixed and random effects. The process of simulating PP, IBI and T was then reiterated until all women had ceased reproduction, which happens necessarily since the probability of parity progression is lower than one. We also set this probability to zero once women reached 60 years old to save computation time in particular simulation scenarios leading to unrealistic life histories (and bad goodness of fit). Note that the maximum recorded age at which a mother gave birth was 55.1 years in our data. For the same reason, we also capped the maximum simulated duration for the interbirth interval to 30 years.

Drawing life-history events from the fit of the model formulas shown above for models 4, 5 and 6 corresponds to simulating the scenario PISH (i.e. all four

hypothetical mechanisms are activated). For simulating other simulation scenarios, we had to fit additional GLMMs derived from models presented above. Specifically, the term twin was dropped from model 4 to deactivate mechanism P (model 8; Supplementary Table 8); the term twin was dropped from model 5 to deactivate mechanism I (model 9; Supplementary Table 9); the term poly(cbind(age, parity), best_order) was dropped from model 6 to deactivate the mechanism S (model 10 and 11; Supplementary Tables 10, 11); and the term (1|maternal_id) was dropped from model 6 to deactivate mechanism H (model 11 and 12; Supplementary Tables 11, 12).

*Testing candidate mechanisms using simulations.* To test how each mechanism or association of mechanisms influenced the relationship between twinning and fertility, we ran simulations under each possible set of activated or inactivated mechanisms. We tested all possible sets and we thus built a total of $4^2 = 16$ simulation scenarios (Supplementary Figs. 3, 4).

For each simulation scenario, we ran simulation replicates (see Supplementary Notes for details and information on the numbers of replicates), then fitted model 3 on the dataset produced by each replicate and extracted the estimate for the slope associated with the term births_total in that model ($\beta^*$). We then consider, in turn, that each simulation scenario may have generated the data. Each simulation scenario is thus considered as a null hypothesis which we aim at testing. Such a test is traditionally referred to as a *goodness-of-fit test*[68]. The result of such a test, a *p*-value, answers the question: what is the probability of obtaining a value equal to, or more extreme than, the statistic of interest, if the null hypothesis were true? The rejection of the null hypothesis by the test (i.e. a *p*-value ≤ 0.05) signifies a rejection of the null hypothesis, and thus, here, the rejection of a simulation scenario which represents a particular mechanism, or combination of mechanisms. In contrast, a large (i.e. non-significant) *p*-value would here denote support for the simulation scenario under consideration.

A first candidate, as a statistic of interest to build our goodness-of-fit test, is the slope $\beta^*$. However, when viewed as a goodness-of-fit test, the direct comparison of the observed and simulated slopes may be conservative when other life-history parameters are fitted to the data. This is because the data tend to be more likely given parameter values fitted to the data than given the actual (unknown) parameter values that generated the data. The goodness-of-fit test is however only guaranteed to provide uniformly distributed *p*-values (a feature necessary for the correctness of any null hypothesis testing) when samples are drawn under the latter parameter values. This is a general issue in statistics which has also been discussed long ago, for example, when the data-generating process is the normal distribution and a Kolmogorov–Smirnov test of goodness-of-fit is applied[69]. We thus designed and validated a specific procedure to correct for such bias while testing each simulation scenario (Supplementary Notes). In the text, we only report outcomes from this unbiased goodness-of-fit test (for details, see Supplementary Notes and Supplementary Table 13).

*Studying the effect of twinning propensity on the number of offspring using simulations.* To study how twinning propensity influences the total number of offspring that mothers produced during their lifetime, we ran two sets of simulations, each with 100 replicates. In the first set, we ran the simulation as described in the section "Simulating the life history of mothers" using the fits of the models associated with the simulation scenario PIS (i.e. fits of models 4, 5 and 12). In the second set, we did the same, except that we modified the intercept of the model predicting twinning events (fit of the model 12) by adding 2.5 to its intercept. We also tried other values, some smaller (e.g. 0.25), some larger (e.g. 5), to make sure that the magnitudes of the change of the intercept did not impact our qualitative statements. For each set of replicates, we extracted the twinning rate, the twinner rate, the mean number of offspring produced and the mean total number of births. We report the means of these metrics, as well as the 95% Central Range from simulation replicates ($CR_{95\%}$), which we directly computed by extracting the corresponding quantiles from the distribution generated by the replicates.

*Realism of the simulations.* We checked that the simulated life history closely matched that of the real mothers represented in our dataset beyond what is captured by the relationship between twinning and fertility. To do so, we compared different metrics related to fertility and twinning between the real and simulated data. We chose to perform this comparison under the simulation scenario PIS since it produces the best goodness-of-fit. The results of this quality check confirm that our simulations represent the reproductive lives of the mothers appropriately (Supplementary Fig. 6).

*Studying the effect of mortality on the number of offspring using simulations.* To account for the fact that not all offspring have the same expected survival, we also applied a survival weight to each simulated offspring before averaging the numbers for a given simulation set (baseline twinning propensity or enhanced, see Results). We used as weights the estimates for the probability of offspring survival between birth and adulthood provided by two publications associated with some of the data we used there. Specifically, following Helle et al.[8], we used a weight of 0.603 for twins, 0.838 for singletons from twinners and 0.815 for singletons from non-twinners. Alternatively, following Haukioja et al.[3], we used a weight of 0.337 for twins and 0.706 for singletons from all mothers.

*Implementation details.* All statistical analyses were performed in R version 4.1[70]. The main R packages we used were spaMM[63] version 3.9.40 for the fit of all the statistical models, boot[71,72] version 1.3-28 for the computation of confidence intervals based on parametric bootstraps, and R6[73] version 2.5.1 for defining the object used to run the simulations. The DESCRIPTION file from our package twinR (see Code availability) lists the additional R packages required for this project (e.g. those used for plotting and data manipulation).

**Reporting summary.** Further information on research design is available in the Nature Research Reporting Summary linked to this article.

## Data availability
The data generated in this study are provided in Supplementary Data 1. They are also available within the supporting R package called twinR (https://github.com/courtiol/twinR) which has been archived within the Zenodo repository: https://doi.org/10.5281/zenodo.6551399.

## Code availability
The R code behind this paper is available within the supporting R package called twinR (https://github.com/courtiol/twinR) which has been archived within the Zenodo repository: https://doi.org/10.5281/zenodo.6551399.

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

## Acknowledgements

The authors thank all the people, too numerous to name, who contributed to generating the datasets used in this study. We also thank Jean Michel Gaillard, Ruth Mace, and Olivia Judson for comments on our manuscript. I.J.R. was supported by Durham University and the German Academic Exchange Service (DAAD). V.L. was supported by the Academy of Finland. I.J.R., A.C. and V.L. also thank the Wissenschaftskolleg zu Berlin which hosted these authors during part of this research. Simulations were run on the DZG server provided by the Leibniz IZW; on the MESO@LR platform, financed by the Occitanie Region, Montpellier Mediterranean Metropole and the University of Montpellier; and on the Montpellier Bioinformatics Biodiversity platform supported by the LabEx CeMEB, an ANR "Investissements d'avenir" program (ANR-10-LABX-04-01). This work was also funded by the Academy of Finland grants no. 317808, 320162, 325857 and 331400 (S.H. and J.E.P.), the Strategic Research Council at the Academy of Finland via the NetResilience consortium grant no. 345185 and 345183 (V.L.), the Kone Foundation grants no. 086809, 088423 and 088423 (S.H. and R.K.), and the Swiss National Science Foundation grants no. 31003A_159462 (E.P. and D.W.). This article was funded by the Deutsche Forschungsgemeinschaft (DFG, German Research Foundation) - project number 491292795.

## Author contributions

I.J.R. and A.C. designed research; E.P., S.H., V.L., R.K., J.P., E.R., G.R.S., C.S., E.V. and D.W. prepared the datasets of each population; I.J.R. compiled the datasets; A.C., I.J.R., F.R. and C.V. analysed data; C.V. and A.C. implemented the numerical simulations; A.C. and C.V. drew the plots and created the tables; F.R. and A.C. contributed new analytic

tools; F.R. drafted the Supplementary Notes; A.C. and I.J.R. wrote the main text; E.P., F.R. and V.L. edited draft versions of this text; and C.V., F.R., E.P., S.H., V.L., R.K., J.P., E.R., G.R.S., C.S., E.V. and D.W. commented on it.

## Funding

## Competing interests
The authors declare no competing interests.
