## [Peer Review File · Nature Communications]

Mothers with higher twinning propensity had lower fertility in pre-industrial EuropeREVIEWER COMMENTS

Reviewer #1 (Remarks to the Author):

Three of the authors (Rickard, Courtiol and Lummaa (RCL)) of this manuscript (ms) penned a comment in 2012 regarding a paper by Robson and Smith (RS) in Proc Roy Soc B in 2011. The 2011 paper presented an analysis of a large data set (>58000 women) from a single natural fertility population showing that women who had birthed twins had lower post-menopausal mortality, shorter interbirth intervals, later ages of last birth and greater lifetime fertility. RS argued that these results were consistent with the maternal heterogeneity hypothesis for dizygotic twinning, a hypothesis that predicts an association between fitness enhancing traits and twinning; although it is unclear whether some of these traits predispose women to the double ovulation that produces twins or makes it more likely that they can successfully bear the costs of twins when double ovulation occurs. RCL's 2012 criticism of the RS paper was that the association between twinning and number of births might simply reflect the fact that if twinning is a random event that can happen in all women with equal probability per delivery (it is not!), then it is inevitable that women that have more births will have a greater chance of having produced twins, leading to the observed association between number of births and twinning. (In the SI of the current ms, the authors convincingly argue that the same criticisms can be applied to the other fitness enhancing traits (excepting post-menopausal mortality) examined by RS.) RS in 2012 replied to the RCL comment with a parity progression analysis of their data which looks at the fraction of women that proceed from one parity to the next. Women that had produced twins were more likely to proceed to the next parity than were those that had not produced twins. RS argued that since such women would have had the same existing number of previous births for each parity progression the RCL criticism was not valid. In the SI of the current ms the authors argue that a similar critique can be applied to parity progression analysis. Specifically, they ask: "do twinning have higher parity progression because they are of higher quality, or are mothers that have higher parity progression more likely to get twins because they give birth more times?" (My changes in wording to improve clarity.) It appears to me that the current ms almost seems like an extension of this debate, using a different analysis of a different data set. Hence, my first suggestion would be to put the SI discussion of the 2011-12 SCL-SR papers in the introduction of the current paper, not in the SI, where it appears almost as an afterthought.

So, what are the new results reported in the current ms. The authors use a data set about half the size of the RS data set coming from multiple sources (perhaps the reason for the large number of authors?) of 18th and 19th century European women to ask how twinning relates to number of births. In their analysis, they attempt to avoid the pitfall ("data aggregation") they feel plagued the RS study. First, they show that without avoiding the aggregation problem women that have produced twins average greater numbers of births, consistent with RS's result. But, when producing twins is defined based on only whether they produced twins at the first birth the number of births for twinning mothers was less than for non-twinning mothers. (But couldn't this result for twinning mothers typically being older?) To estimate overall twinning propensity and compare that to number of births they estimated "the relationship between per birth twinning probability and total births based on the twinning status for each birth." (I am not actually sure what this means as the phrase "based on the twinning status for each birth" made the sentence confusing. The authors should clarify this.) Once again, the relationship was negative: greater twinning propensity, fewer births. (But was maternal age standardized?) Thus, these results are taken as evidence against the maternal heterogeneity hypothesis. There is more to the maternal heterogeneity hypothesis than number of births, so it is unclear to me why the authors didn't simply use similar analyses to examine the relationship between twinning propensity and the other traits indicative of maternal heterogeneity: post-menopausal mortality, shorter interbirth intervals, later ages of last birth. Instead, the authors explore the causes of the negative relationship between twinning propensity and total births by looking in their data for evidence of four causal mechanisms potentially responsible for the negative relationship between twinning and total births: parity progression (change in probability of subsequent reproduction), interbirth intervals (change in time to subsequent birth following a twin birth, but not overall?),

maternal age (which may covary with twinning and total number of births), heterogeneity among mothers (random differences between mothers in twinning and number of births). To do this they created statistical models to estimate from the data set the effects (polynomial coefficients?) associated with these mechanisms. (I am not totally competent to judge the validity of these statistical models, but they seem reasonable.) Based on these effects they conducted simulations of reproducing females using all possible (16) combinations of the effects to see which produced the best (goodness of) fit to the patterns observed in the data vis a vis the relationship between twinning and number of births. From these results, they concluded that the negative relationship between per birth twinning and number of births occurred because (a) when twins were produced it often happened when mothers were older than average and they had fewer subsequent births, and (b) after producing twins, mothers were more likely to cease reproduction. They also show that although the relationship between twinning propensity and number of births is negative, women with greater propensity to produce twins have more offspring. Of course, it would be more useful to know something about the survival of these offspring to reproduction.

Approximately 25% of the discussion involves the above results. These are the sections: "Negative associations between twinning and total births," "Two mechanisms responsible for the association between twinning and total births," "The presence of stochasticity in life histories," and "The presence of maternal heterogeneity." The remaining sections (three!) are, in my opinion, mostly handwaving, having little direct connection to the data presented and analyzed. Nothing in the current ms sheds much light on differences between populations in twinning, for example, and the other two sections are about issues that can arise from the "aggregation of data" problem. If aggregation is what the authors wanted the paper to be about, they should have explicitly raised this issue in the introduction. In fact, at this point I'm not sure what the main point of the paper is. Is it maternal heterogeneity and twinning, or the dangers of data aggregation?

It seems to me that the value of this paper is that it adds to the list of postnatal costs and benefits associated with birthing twins; mothers of twins start reproducing later and after producing twins they are less likely to produce additional offspring (Wouldn't this be because older women are more likely to produce twins and less likely to produce subsequent offspring?), even if overall they produce more babies. This information is valuable for attempts to investigate the evolution of twinning (see for example Hazel et al. 2020, where variation maternal survival during childbirth, prenatal survival and postnatal survival to reproduction are used to understand how selection operates on the twinning). So overall, with respect to selection on twinning this paper does not break much new ground. This is partly because numbers of births are only part of the picture. For example, it is now well known that a massive amount of human mortality of relevance to understanding why twinning evolved occurs prenatally, most very early. Indeed, twinning rate at birth reflects both the rate of double ovulation and the prenatal survival rate (Hazel et al. 2020). Both double ovulation and prenatal survival are age dependent and lead to the results shown in Fig 3C (compare this with Fig 1 in Hazel et al. 2020); this is a result that seems to have been lost on the authors of the current ms.

In summary, because this ms is so narrowly, but schizophrenically focused on the relationship between twinning and number of births on one hand and methodological issues associated with data aggregation on the other, I do not see it rising to a level of importance appropriate for publication in Nature Communications.

Reviewer #2 (Remarks to the Author):

Comments on Rickard et al. Determinants of twinning..

This is a really thorough investigation of the determinants of twinning in several large historical databases from northern European countries in the 18th century. It includes individual level and multilevel statistical analysis, and simulation models. They find support for the effects of twinning on

parity progression and effects of age on risk of twinning, and conclude that these effects are enough to explain positive correlations between twinning and reproductive success. They do not think there are physiological or intrinsic differences between mothers of twins and others, as other authors working on the fitness of twinning have concluded and argue their conclusions may be erroneous. This results is then backed up by a simulation study that identifies these two hypotheses as by far the most important determinants in generating the observed relationships. This is certainly the most thorough analysis of twinning that I am aware of, and a great contribution to the literature. They go on to discuss a number of other contexts in which similar statistical comparisons may have reached the wrong conclusion.

Points for consideration

Physiological differences:

Whilst the results here reject the notion of intrinsic differences between twin mothers, if I remember rightly Sear et al 2001 found that twin mothers in a Gambian natural fertility population were physiologically different, such as a bit taller and higher weight for height as teenagers, than non-twin mothers. The data used here does not allow that to be tested as far as I can see, being demographic data only. But I think this finding should be discussed. It suggests that there are intrinsic differences which are important in that population, and despite that, that many of the results found are due to mothers decision-making to ameliorate costs of reproduction.

Population variation:

Is it possible that twinning in northern Europe is not under the same selective pressure as twinning in Africa. I think twinning is far more common in west African populations than in European ones. This might be because the costs of multiple births are lower in Africa and therefore there may be selection in favour of twinning that is perhaps not found in northern European groups.

Definition of twin mothers by first birth:

It is quite well known in high fertility human populations, and I think even by farmers of commercial livestock for that matter, that a first birth that is twins can be very detrimental to a young mothers health to the extent that it is often accompanied by infanticide or fostering (sorry no reference here). So, it is possible that there could be active selection against first births being twins as first births, making it a slightly unreliable way to define twin mothers.

Ruth Mace

Reviewer #3 (Remarks to the Author):

This paper was a fascinating look at twinning, using robust historical datasets. The authors convincingly demonstrate that we should all be studying per-birth twinning likelihood, because mothers who have more births are of course more likely to have twins. They use simulations to disentangle potential drivers of their observed effects. A fascinating conclusion is that the traditional hypothesis that some "super moms" have way more births and way more twins is not supported. This paper was a pleasure to read, was clearly written, and raises many new research questions. Here, I include several broad and specific comments.

Broad comments:

FERTILITY

How are you defining fertility? I have nearly always seen fertility defined as "the number of offspring a mother has in her life." You show through simulation that twinning increases the number of offspring, but you also show that twinning decreases birth events. It was jarring to read the following quotes in quick succession:

- 324-6: the total number of offspring mothers produced during their lifetime did increase with twinning propensity, despite the reduction that the latter imposed on total births.
 - Followed immediately by "it is twinning that impacts fertility and that such an impact is negative."
- The way you talk about and define fertility should be carefully defined and adjusted throughout based on conventions in the literature.

RELATEDNESS:

- Are you able to account at all for family structure between mothers? If impossible this should be noted.

MORTALITY of TWINS vs. SINGLETONS

- Something the paper is broadly missing, and that which is related to the # offspring vs. # births comment above, is the mortality rates of twins.
- The authors should find estimates from the human biology literature and apply them in their paper to come up with mortality-adjusted estimates of fertility (# living offspring at age 1, say) between twinning and non twinning. (or low twinning and high twinning)
- The non-human primate literature may be useful here, because many papers on non-human primates have documented strong effects whereby singletons have far higher survivorship than twins, a difference that arises very soon after birth (that is, twins have mortality right near birth) – see, e.g.:
 - ♣ McCoy et al. (2019) "A comparative study of litter size and sex composition in a large dataset of callitrichine monkeys." *American journal of primatology*;
 - ♣ Ward et al. (2014). Twinning and survivorship of captive common marmosets (*Callithrix jacchus*) and cotton-top tamarins (*Saguinus oedipus*). *Journal of the American Association for Laboratory Animal Science*;
 - ♣ Harris, R. Alan, et al. (2014) "Evolutionary genetics and implications of small size and twinning in callitrichine primates." *Proceedings of the National Academy of Sciences*

MATERNAL AGE

In general, why do you not include maternal age in all of the models? It is known to strongly influence twinning rates.

DATA and ANALYSIS

- It is useful in studies with lots of data to, at some point, see the real data itself rather than just the results of models and simulations. For example, to see the distribution of # of births and # of offspring in the data. Could this be included, and raw data plotted where possible?
- In general, I think too much about the models is in the supplemental (essentially hidden from the casual reader's view)—model predictors for key findings should be included in the figure captions and text, such as when maternal age is considered (or not).

EVOLUTIONARY CONTEXT OF TWINNING

Twinning is a very cool and unusual evolutionary phenomenon, and I would like to see reference to key

LINE COMMENTS

59-60: repeated words

Figure 1:

- Include statistics in the caption for the reader who likes to look at figures
- Include the predictors and summary stats of the GLMM models

- C-D: Maternal age is a strong predictor of twinning potential. Was this included in the model of twinning-at-first-birth? This could be a very simple reason why twinning at first birth decreases lifelong births. Why not just look at per-birth twinning likelihood rather than twinning at first birth likelihood?

Figure 2

- Include statistics
- Include summary stats and predictors of GLMM model
- Should the axis include 0? This is a little bit deceptive, particularly since it is already a strange axis (although you kindly explain why in the caption)

151-153: This raises the question- why study first birth at all when you could instead study per birth twinning potential? (I am suggesting you justify, rather than exclude, this analysis)? I can imagine good reasons to do so. One might be to carefully consider maternal age without other confounding factors such as parity. However, such a decision sort of needs to be justified given that you have a more comprehensive approach available.

154-155: "this analysis confirms that the relationship between twinning propensity and total births is not positive at the level of births."

I thought this sentence could be more clearly worded as something like (no need to use this exact wording): "Per-birth twinning propensity is negatively related to total births. That is, mothers who were more likely to have twins in a given birth event were not likely to have more births overall."

158-161: You describe the different between a mother of 1 and mother of 18, and the confidence intervals nearly overlap—could you include the same sentence for mothers + and - one or two standard deviations from the median # of births so we can better understand the scale of the effect?

181-182: "mothers were indeed less likely to reproduce following the delivery of twins than following the delivery of a singleton" Were mothers slower to have their next birth after having twins? Also, what does "the probability of parity progression" mean in simple terms—the likelihood that mothers have any more births, period? This should be described here and in the fig., caption.

178-206: My stylistic preference would be to avoid "mechanism A, B, C phrasing" which I have to go back to refer to, and instead replace these with the actual name of the mechanisms (interbirth intervals, parity, maternal age). I would prefer this change to be made throughout the whole document.

Figure 4:

- Figure 4, could you add a legend for what each letter refers to? Also, I would find it easier to parse if it were oriented horizontally rather than in a circle, but I understand if you prefer this design choice.
- It is important to note that not rejecting the null model does not mean accepting the null model

208: "heterogeneity between mothers" is ambiguous. I would suggest more precise phrasing, such as "mothers who twin often also are more fertile overall"

274-275: "the impact of twinning events on parity progression and the effect of the schedule of reproduction" – can this be replaced with a less jargony "parity effects and maternal age effects", or is that not precise enough?

274: A simple way to understand the impact of maternal age is look only at first birth, whether or not it is a twin, and include age as a predictor. If I am understanding your paper correctly you didn't do that model.

313-327: Any conclusion about fitness drawn from this need to incorporate statistics from the literature about twin mortality rates compared to singleton mortality rates.

339: "misconceived study designs." Since this is the paragraph most likely to be read, after the abstract that is, say (in a phrase) in what way the designs were misconceived.

360-366: You contrast "reduced reproductive capacity" with "foregoing further reproduction." What actual mechanism underlies the observed "foregoing"? Is it reduced capacity? Choice? Birth control? Reduced capacity due to a busy life caring for twins? Reduced biophysical capacity? Etc.

368: Here, the reader has forgotten what Mechanism A is (if they are tired and forgetful like me, at least).

403: between populations? What between-population analyses did you do?

406: You have not adjusted for mortality in order to conclude: "the additional children brought by a twin birth more than compensated for the reproductive cost that twinning imposes on mothers."

414-422: including much recent work on primates, including:

423-462: great explanation!

476-482: can you cite some of the studies I question here, both to demonstrate that it is a real problem and draw attention to this issue?

800-802: wonderful to have the R Package available.

Please find below a summary of the four (i - iv) important changes we made in the manuscript, followed by point-by-point responses to the reviewers' comments.

Noticeable changes to the manuscript

1. Revised framing

The main criticism from Reviewer 1 concerned the framing and organisation of our paper. In particular, Reviewer 1 pointed out two general issues: (i) some key biological background was only present in the supplementary material (*"my first suggestion would be to put the SI discussion of the 2011-12 RCL-RS papers in the introduction"*); (ii) the introduction failed to mention the aggregation fallacy and to connect it to twinning (*"They should have explicitly raised this issue in the introduction"*). The result was a manuscript that the reviewer described as *"schizophrenically focused on the relationship between twinning and number of births on one hand and methodological issues associated with data aggregation on the other"*.

We agree with this criticism and have thus revised and extended our introduction, and rewritten the discussion, so as to address these matters. We believe that the revised framing benefits the manuscript in the following ways: (i) the goal of the paper becomes clear and is now broader: we state explicitly that we are interested in the association between twinning and fertility beyond the heterogeneity hypothesis; (ii) the aggregation fallacy is introduced as part of the biological story and no longer as a disjointed topic that first shows up in the discussion; (iii) the RCL-RS debate is now referred to in the main text, with no loss of generality with respect to the scope of the introduction; (iv) the novelty of the paper is much better expressed (this is the first study assessing the biology of the relationship between fertility and twinning – comprehensively and with no bias caused by an aggregation fallacy).

In more detail: our revised introduction draws in elements from both the discussion and supplementary information (SI) from the previous draft, and also brings in new materials that help explain the relevance of the study. We also mention key elements from the debate 2011-12 RCL-RS formerly presented in SI; however, we remained brief on this topic since our framing is now more general than before. The new organisation of the introduction tightly links our research question (*what is the relationship between twinning and intrinsic fertility?*) to both its implications (theoretical and medical) and the issue of data aggregation (which distorts how one interprets the relationship). With this set out early on, it became possible for us to write a discussion that is both shorter and more connected to our results, avoiding an important charge made by Reviewer 1 (*"The remaining sections (three!) are, in my opinion, mostly handwaving, having little direct connection to the data presented and analyzed."*). See also point 4 below, for why our discussion is now much improved.

2. Improved explanation of when and why maternal age is/isn't controlled for

Maternal age is the clearest predictor of the variation in twinning rate and it is thus a crucial element to consider during any analysis of twinning. Reviewer 1 and Reviewer 3 did not

understand exactly in which statistical analysis we controlled for maternal age and in which
ones we did not, nor why we did not always apply the same strategy.

To clarify our methodology, we are now being explicit about when and why we controlled for
50 maternal age or not (see lines 209-238). We have also added a new figure which clearly
illustrates how maternal age is accounted for in each mechanism (new figure 3).

In more detail: we present four (previously five) main series of analyses in the paper. The
54 first aims at replicating previous published findings to show that our dataset is consistent
with others. For this first analysis, it does not actually matter if maternal age is controlled for
or not, since maternal age does not qualitatively affect the result, and previous publications
were not consistent about applying such a control. Nor is it obvious how maternal age should
be controlled for in the context of a lifetime measure. The second series of analyses – of
twinning status at first birth – is where the reviewers did not understand why we did not
control for maternal age. The point is now moot since we deleted this particular analysis (see
point 3 below). In what is now presented as the second analysis (formerly third), we want to
document the raw relationship between fertility and twinning at the level of births. Here, it is
crucial that we do not control for maternal age because we deliberately want the relationship
to be influenced by this trait as well as by all other factors that may impact the relationship.
We do it this way because the third series of analyses (formerly fourth) aims to reproduce
the raw relationship between fertility and twinning by simulating, and comparing the roles of
different biological mechanisms. All such mechanisms do include the role of maternal age,
but we distinguish through which life history traits maternal age may actually exert its
influence. It is thus to better characterise the effect of maternal age in the third series of
analyses that we omit it in the second series. The last series of analyses, in which we
compute the reproductive success of women with or without an increased twinning
propensity, does consider the effect of maternal age on all three life history traits modelled.
We implemented this because the previous series of analyses identified this biological
scenario as the one fitting the data the best.

3. Problematic and unnecessary analysis removed

All three reviewers expressed concerns about the analysis of the twinning status at first birth.
In retrospect this analysis was not necessary for our argument and we have thus decided to
drop it altogether.

In more detail: the idea of performing an analysis on the twinning status at first birth was to
show that even without any sophisticated treatment, once the data are no longer
aggregated, the relationship between fertility and twinning is negative. In retrospect, the
benefit can only be understood after reading the section about the goodness-of-fit analysis.
We have thus concluded that the analysis of first births is one step too many in our
argument: this particular analysis does not add anything substantial to our conclusion, while
it raises a series of questions calling for justifications that would inflate the manuscript with
peripheral elements.

4. Role of mortality included

Reviewers 1 and 3 argued that analyses failing to account for the difference in mortality
between twins and singleton offspring do not quite deliver when one wants to discuss natural
selection acting on twinning. They are correct; to solve the issue, we followed reviewer 3's
proposition to rely on published mortality estimates for singleton and twin offspring.

In more detail: we were able to retrieve mortality estimates for the particular populations we
sampled since some had been published. The revised analysis shows twinning propensity
can exert either a positive or negative effect on the total number of surviving offspring,
depending on the exact mortality rates considered for twins and singleton. This is a
fascinating result, which provides good material for us to discuss the different hypotheses
about the evolution of twinning. This new result (see lines 440-451) and the associated
discussion (lines 543-566) should particularly please Reviewer 1, who was unimpressed by
our paper. In particular, the reviewer's objection that "*Nothing in the current ms sheds much
light on differences between populations in twinning, for example*" no longer applies. The
new result also explains why we amended the original title of our manuscript.

## Point-by-point responses to the reviewers' comments

**Reviewer #1 (Remarks to the Author):**

*Three of the authors (Rickard, Courtiol and Lummaa (RCL)) of this manuscript (ms) penned a
comment in 2012 regarding a paper by Robson and Smith (RS) in Proc Roy Soc B in 2011. The 2011
paper presented an analysis of a large data set (>58000 women) from a single natural fertility
population showing that women who had birthed twins had lower post-menopausal mortality, shorter
interbirth intervals, later ages of last birth and greater lifetime fertility. RS argued that these results
were consistent with the maternal heterogeneity hypothesis for dizygotic twinning, a hypothesis that
predicts an association between fitness enhancing traits and twinning; although it is unclear whether
some of these traits predispose women to the double ovulation that produces twins or makes it more
likely that they can successfully bear the costs of twins when double ovulation occurs. RCL's 2012
criticism of the RS paper was that the association between twinning and number of births might
simply reflect the fact that if twinning is a random event that can happen in all women with equal
probability per delivery (it is not!), then it is inevitable that women that have more births will have a
greater chance of having produced twins, leading to the observed association between number of
128 births and twinning. (In the SI of the current ms, the authors convincingly argue that the same
criticisms can be applied to the other fitness enhancing traits (excepting post-menopausal mortality)
examined by RS.) RS in 2012 replied to the RCL comment with a parity progression analysis of their
data which looks at the fraction of women that proceed from one parity to the next. Women that had
132 produced twins were more likely to proceed to the next parity than were those that had not produced
twins. RS argued that since such women would have had the same existing number of previous births
for each parity progression the RCL criticism was not valid. In the SI of the current ms the
authors argue that a similar critique can be applied to parity progression analysis. Specifically, they
ask: "do twinners have higher parity progression because they are of higher quality, or are mothers
that have higher parity progression more likely to get twins because they give birth more times?" (My*

*changes in wording to improve clarity.) It appears to me that the current ms almost seems like an*
*extension of this debate, using a different analysis of a different data set. Hence, my first suggestion*
*would be to put the SI discussion of the 2011-12 SCL-SR papers in the introduction of the current*
*paper, not in the SI, where it appears almost as an afterthought.*

We thank Reviewer 1 for taking the time to recap the RCL-RS discussion and we agree with
144 the summary sketched above. We have followed the reviewer's suggestion to bring some of
the elements from the supplementary material into the introduction (lines 117-125).

However, we remained brief as we did not want to dwell on this particular controversy.

The point of the current manuscript is to thoroughly assess how different biological
mechanisms contribute to shaping the relationship between twinning and fertility –
150 something neither RS nor anyone else attempted before. This should now be clear from the
outset (lines 58-59), and the revised discussion should also better demonstrate why a proper
understanding of the relationship between twinning and fertility has important implications
(lines 480-523).

Another novel aspect of our paper is that we deploy a methodology that does not risk an
156 ecological fallacy. This contrasts with previous approaches investigating the relationship
between twinning and fertility—whether from RS or from other authors (including many of
158 us!)—which all suffered from the same design flaw (births were aggregated before twinning
was analysed, leading to the neglect of variation in the exposure to the risk of twinning; lines
95-136).

That our paper is not "*an extension of [the RCL-RS discussion], using a different analysis of*
a different data set" should now be clearer than before since we have revised and
164 broadened the framing of the paper. In particular, we have no interest in debating further
whether or not the particular conclusions of the RS papers are robust to the pitfall caused by
166 the aggregation fallacy. We have now clarified that we remain agnostic about that since we
did not use their data – see lines 507-510. We have also clarified that "super mothers" may
be said to exist in a sense, but in our data they bear no role on the relationship between
twinning and fertility (lines 504-507). All these considerations finally led us to remove the
170 particular section on RCL-RS present in the SI, since the key ideas are now in the main text
and, in the light of the new framing, it no longer seems relevant to focus on these studies in
such depth.

*So, what are the new results reported in the current ms.*

Most results presented in our paper are novel. First of all, the finding that the relationship
between twinning and fertility changes direction due to aggregation is an important novel
result (lines 187-195, figure 2) with a myriad of practical implications (lines 469-566). To our
knowledge, it also marks the first finding of a Simpson's paradox caused by the aggregation
of events within the lives of individuals (lines 464-468). Second, the results on the
comparison of different biological mechanisms that contribute to shaping the relationship
between twinning and fertility are also novel (lines 368-384, figure 5) and clarify our
understanding of the biology of twinning. It is true that most of the mechanisms we proposed

have been studied before, but they had not been compared. Finally, the finding that the
mortality costs of twinning may shape natural selection more than its fertility benefits (lines
193-198 & 377-381 vs 440-451) is also a novel result with interesting consequences for
understanding the evolution of twinning (lines 524-556).

*The authors use a data set about half the size of the RS data set coming from multiple sources
(perhaps the reason for the large number of authors?) of 18th and 19th century European women to
ask how twinning relates to number of births. In their analysis, they attempt to avoid the pitfall (“data
aggregation”) they feel plagued the RS study. First, they show that without avoiding the aggregation
problem women that have produced twins average greater numbers of births, consistent with RS's
result.*

Yes, this is a correct summary of what we did, but we want to remark here about two details
that are important to keep in mind while assessing our work. First, as mentioned above, the
198 results we found when aggregating data not only replicate the RS results but also the results
of many other studies, including several previous ones performed on the populations we
have examined (see lines 170-176, 457-464). Moreover, we find the comparison of the size
of the two datasets to be misleading: since RS used aggregated data throughout, the sample
size relevant for their study is the number of mothers. In the present paper, however, the
final analyses are all produced on non-aggregated data, meaning that the sample size for
our study is given by the number of births. With this statistical understanding in mind, the
size of our dataset is not half but twice that of the one used by RS. We see no reason to
206 brag about this in the text since both datasets are highly valuable (lines 138-148), but we
object to the comparison made by Reviewer 1.

Reviewer 1 continues, remarking:

*But, when producing twins is defined based on only whether they produced twins at the first birth the
212 number of births for twinning mothers was less than for non-twinning mothers. (But couldn't this
result for twinning mothers typically being older?)*

The question raised by the reviewer is precisely one of the questions our manuscript
addresses. We demonstrate that maternal age is indeed a driver of the relationship between
fertility and twinning rate, but not the only one (lines 211-220).

The comment reveals two issues about the previous version of our paper: (i) that it was not
clear when and why some analyses controlled for maternal age while others did not, and (ii)
that it was not obvious how to interpret the outcome of the analysis on the twinning status at
222 first birth. As discussed above (point #2 and #3 in section "Noticeable changes to in the
manuscript"), we have made general changes to the manuscript to tackle these two issues.

*To estimate overall twinning propensity and compare that to number of births they estimated “the
226 relationship between per birth twinning probability and total births based on the twinning status for
each birth.” (I am not actually sure what this means as the ph[r]ase “based on the twinning status for
each birth” made the sentence confusing. The authors should clarify this.)*

We have now amended this sentence to make our intended meaning clearer (see lines 191-
192). The point is that we analyse the twinning outcome at the level of each birth rather than
at the level of each mother.

*Once again, the relationship was negative: greater twinning propensity, fewer births. (But was
maternal age standardized?)*

No, the relationship depicted in figure 2 is not controlled for the variation in maternal age, but
this is now stated explicitly in the manuscript (lines 212-213). See discussion above (point #3
in section "Noticeable changes to the manuscript").

*Thus, these results are taken as evidence against the maternal heterogeneity hypothesis. There is
more to the maternal heterogeneity hypothesis than number of births, so it is unclear to me why the
authors didn't simply use similar analyses to examine the relationship between twinning propensity
and the other traits indicative of maternal heterogeneity: post-menopausal mortality, shorter
interbirth intervals, later ages of last birth. Instead, the authors explore the causes of the negative
relationship between twinning propensity and total births by looking in their data for evidence of four
causal mechanisms potentially responsible for the negative relationship between twinning and total
248 births: parity progression (change in probability of subsequent reproduction), interbirth intervals
(change in time to subsequent birth following a twin birth, but not overall?), maternal age (which may
covary with twinning and total number of births), heterogeneity among mothers (random differences
between mothers in twinning and number of births).*

We agree that we could have looked at the effect of maternal heterogeneity on many
reproductive outcomes other than twinning, but we favoured a thorough analysis of a single
key relationship rather than a superficial exploration of several relationships less relevant to
256 understanding the relationship between twinning and fertility and thus its evolution. Yet, by
focussing on the relationship between twinning and the number of births, we actually capture
everything that influences realised fertility, such as the effects of interbirth intervals, ages of
last birth, and onset of menopause. We also discuss why the issues we bring to light are
260 relevant to previous research on other traits (lines 512-523). Finally, we do document the
presence of maternal heterogeneity for each life history trait investigated (parity progression,
interbirth intervals and twinning rate) and discuss the heterogeneity in both twinning and
fertility between mothers. What our study shows, however, is that such heterogeneity does
not drive the relationship between twinning and fertility (lines 368-384, 497-510) and is thus
unlikely to be key for understanding the evolution of twinning and variation in twinning rates
(lines 524-527).

*To do this they created statistical models to estimate from the data set the effects (polynomial
coefficients?) associated with these mechanisms. (I am not totally competent to judge the validity of
270 these statistical models, but they seem reasonable.)*

Yes, the polynomial coefficients characterise the effect of predictors on particular life history
traits relevant for understanding the relationship between fertility and twinning. We used
polynomials because relationships are not well approximated by linear relationships. The
interpretation of polynomial coefficients is arguably more difficult than that of the slope of

276 linear functions, but we have plotted the fitted relationship in figure 4, which circumvents this
difficulty.

*Based on these effects they conducted simulations of reproducing females using all possible (16)
combinations of the effects to see which produced the best (goodness of) fit to the patterns observed in
the data vis a vis the relationship between twinning and number of births. From these results, they
concluded that the negative relationship between per birth twinning and number of births occurred
because (a) when twins were produced it often happened when mothers were older than average and
284 they had fewer subsequent births, and (b) after producing twins, mothers were more likely to cease
reproduction. They also show that although the relationship between twinning propensity and number
of births is negative, women with greater propensity to produce twins have more offspring.*

Yes, this is a correct summary of what we wrote.

*Of course, it would be more useful to know something about the survival of these offspring to
reproduction.*

We agree. This is why we have now taken into consideration the fact that singleton and twin
offspring do exhibit different survival rates (see point #4 in section "Noticeable changes to
the manuscript").

*Approximately 25% of the discussion involves the above results. These are the sections: "Negative
associations between twinning and total births," "Two mechanisms responsible for the association
between twinning and total births," "The presence of stochasticity in life histories," and "The
300 presence of maternal heterogeneity." The remaining sections (three!) are, in my opinion, mostly
handwaving, having little direct connection to the data presented and analyzed.*

We have completely rewritten the discussion in view of this (fair) criticism (see point #1 in
section "Noticeable changes to the manuscript").

*Nothing in the current ms sheds much light on differences between populations in twinning, for
example, and the other two sections are about issues that can arise from the "aggregation of data"
problem. If aggregation is what the authors wanted the paper to be about, they should have explicitly
raised this issue in the introduction. In fact, at this point I'm not sure what the main point of the paper
is. Is it maternal heterogeneity and twinning, or the dangers of data aggregation?*

We agree that the previous version of our manuscript was a little light on the matter of
differences between populations. We thus hope that the reviewer will be pleased to learn
that (i) we now provide more detailed results on how the relationship between twinning and
fertility may differ between populations (lines 195-198), and that (ii) the additional
316 consideration of mortality in our analyses (see point #4 in section "Noticeable changes
performed in the manuscript") reveals that the cost of twinning did change between
318 populations, which we consider a plausible reason why twinning rate varies (lines 543-556).

Concerning what the paper is about, and the link between twinning and data aggregation,
the revised framing of the paper should now make clear that what we consider to be the

322 main focus of the paper is why twinning varies (lines 58-59), and that it is necessary to tackle
the obstacle of data aggregation to get there (see point #1 above in section "Noticeable
changes to the manuscript"). We agree with the reviewer that it was a mistake not to
mention the aggregation fallacy in sufficient detail in the previous version of our introduction.

Failing to recognise the true danger brought by aggregation is a practice that has resulted in
papers making many false claims about many important topics. For some research
questions, the danger of the aggregation fallacy is more widely recognised and often tackled
accurately. This is not, however, the case for studies of twinning, which is why we spend
some time discussing the matter – a matter with impacts not only on eco-evolutionary
narratives (lines 524-556) but also on the future of medical research for fertility treatment, as
well as on public health policies (see lines 480-495).

*It seems to me that the value of this paper is that it adds to the list of postnatal costs and benefits
associated with birthing twins; mothers of twins start reproducing later and after producing twins
they are less likely to produce additional offspring (Wouldn't this be because older women are more
likely to produce twins and less likely to produce subsequent offspring?), even if overall they produce
more babies. This information is valuable for attempts to investigate the evolution of twinning (see for
340 example Hazel et al. 2020, where variation maternal survival during childbirth, prenatal survival and
postnatal survival to reproduction are used to understand how selection operates on the twinning).*

We are happy to read that Reviewer 1 considers some aspects of our study as "valuable".
Concerning the question "*Wouldn't this [the decline in parity progression after the birth of
twins] be because older women are more likely to produce twins and less likely to produce
subsequent offspring?*": No it is not; our results clearly show that the effect of maternal age
on the twinning rate (mechanism S1, formerly C1) and the effect of the production of twins
on the next parity progression (mechanism P, formerly A) are both at play but distinct. The
goal of our goodness-of-fit analysis, which compares the support for different combinations
of four biological mechanisms, is precisely designed to disentangle their relative effects. This
is now better explained throughout the Results section (e.g. lines 211-220, 349-366).

So overall, with respect to selection on twinning this paper does not break much new ground.

We hope that the revised version of our paper will address this general criticism and we
recall that other reviewers assessed our study differently (e.g. Reviewer 2 wrote "*This is
certainly the most thorough analysis of twinning that I am aware of, and a great contribution
to the literature*" and reviewer 3 wrote "*This paper was a fascinating look at twinning*").

*This is partly because numbers of births are only part of the picture. For example, it is now well
known that a massive amount of human mortality of relevance to understanding why twinning evolved
occurs prenatally, most very early.*

We agree. This is why we have now taken into consideration the fact that singleton and twin
offspring do exhibit different survival rates (see point #4 in section "Noticeable changes to
the manuscript").

*Indeed, twinning rate at birth reflects both the rate of double ovulation and the prenatal survival rate*
*(Hazel et al. 2020). Both double ovulation and prenatal survival are age dependent and lead to the*
*results shown in Fig 3C (compare this with Fig 1 in Hazel et al. 2020); this is a result that seems to*
*have been lost on the authors of the current ms.*

The reviewer is right that the excellent work of Hazel et al. 2020 and our results are perfectly
compatible (the figure 3C – now figure 4C – in our paper resembles figure 1 from Hazel et al.
2020), but this is not something that we either failed to notice, or failed to discuss. We had
mentioned the study and linked our results to it (we had written about figure 3C "*It is thought*
*to result from women being more likely to undergo double ovulation as they age, while*
*simultaneously experiencing an increasing risk of prenatal mortality for the embryos – an*
*idea called the insurance ova hypothesis"* and cited Hazel et al. 2020 as well as other
papers) and this is still the case in this revised version (see lines 524-542). In fact, we see
Hazel et al.'s study and ours as being complementary views on what drives twinning
because the two studies focus on different levels of the biological explanation. Our study
identifies the effect of age on twinning (mechanism S) as one of the two main drivers of the
relationship between maternal fertility and twinning (see lines 382-384). Hazel et al. provide
a convincing physiological explanation for why this is the case.

*In summary, because this ms is so narrowly, but schizophrenically focused on the relationship*
*between twinning and number of births on one hand and methodological issues associated with data*
*aggregation on the other, I do not see it rising to a level of importance appropriate for publication in*
*Nature Communications.*

We hope that the revised version of our paper will be perceived more positively by this
reviewer.

**Reviewer #2 (Remarks to the Author):**

*Comments on Rickard et al. Determinants of twinning..*

*This is a really thorough investigation of the determinants of twinning in several large historical*
*databases from northern European countries in the 18th century. It includes individual level and*
*multilevel statistical analysis, and simulation models. They find support for the effects of twinning on*
*parity progression and effects of age on risk of twinning, and conclude that these effects are enough*
*to explain positive correlations between twinning and reproductive success. They do not think there*
*are physiological or intrinsic differences between mothers of twins and others, as other authors*
*working on the fitness of twinning have concluded and argue their conclusions may be erroneous.*
This results is then backed up by a simulation study that identifies these two hypotheses as by far the
most important determinants in generating the observed relationships. This is certainly the most
thorough analysis of twinning that I am aware of, and a great contribution to the literature.
They go on to discuss a number of other contexts in which similar statistical comparisons may have
reached the wrong conclusion.

This is a correct summary of our study and we are very pleased to read that the reviewer
considers our paper to be "*the most thorough analysis of twinning*" and "*a great contribution*
*to the literature*".

*Points for consideration*

*Physiological differences:*

*Whilst the results here reject the notion of intrinsic differences between twin mothers, if I remember*
*rightly Sear et al 2001 found that twin mothers in a Gambian natural fertility population were*
*physiologically different, such as a bit taller and higher weight for height as teenagers, than non-twin*
*mothers. The data used here does not allow that to be tested as far as I can see, being demographic*
*data only. But I think this finding should be discussed. It suggests that there are intrinsic differences*
*which are important in that population, and despite that, that many of the results found are due to*
*mothers decision-making to ameliorate costs of reproduction.*

The study by Sear et al. 2001 is an excellent and rich study which we cite. In it, the authors
studied the relationships between many characteristics of mothers and their twinning status,
including anthropometric measurements. Yet, they actually found non-significant differences
for height, weight and BMI between twinning and non-twinning. They do describe tendencies
(twinning being slightly taller, heavier and with slightly higher BMI), but the p-values were
large enough not to be ambiguous ($p = 0.251$, $p = 0.604$ and $p = 0.510$, respectively). The
only significant results reported for anthropometric traits is that the weight-for-height of girls
who later became twin mothers was a little larger than that of girls who did not.

Unfortunately, we do not know if we can trust this result (beyond the issue of small sample
size and multiple testing) because as the authors wrote: "*[they] compared the*
*anthropometric status of twin and singleton mothers*", and by focussing on the mother level
and not on the level of births, this particular study fell right into the aggregation trap (like all
other research investigating the benefits of twinning). They did actually use multilevel
regression models (the right tool to analyse non-aggregated data), but only insofar as to
account for repeated measurements performed on weight; unfortunately not for accounting
for differences between individuals in their exposure to the risk of twinning.

To better understand the issue, and assuming that the result on weight-for-height is genuine,
it suffices to consider for example that weight-for-height is related to fertility. In this case,
individuals with a higher weight-for-height would go on to reproduce more than others. The
issue is that such an effect on fertility would on its own increase the exposure to the risk of
twinning. If that were true, then the effect of weight-for-height on fertility would generate an
association between twinning and weight-for-height (when twinning and non-twinning are
compared; i.e. when aggregated data are analysed) without the anthropometric trait actually
modifying the twinning probability at each birth.

In sum, the paper by Sear et al. 2001 does not distinguish between a direct effect of
variables (anthropometric or not) onto the probability of twinning, from the indirect effect of

variables onto the exposure to the risk of twinning. Yet, only the former is relevant to
understand the evolution of the twinning propensity.

Just to be extra cautious: we are not saying that anthropometry has no direct relation to
twinning, we are simply saying that the studies that claim it is the case, such as that from
Sear et al. 2001, fail to demonstrate it.

Since this discussion illustrates slightly differently the very same problem we discuss
throughout our entire manuscript, we have now added a small paragraph about it in our
Discussion (lines 512-523).

*Population variation:*

*Is it possible that twinning in northern Europe is not under the same selective pressure as twinning in*
*Africa. I think twinning is far more common in west African populations than in European ones. This*
*might be because the costs of multiple births are lower in Africa and therefore there may be selection*
*in favour of twinning that is perhaps not found in northern European groups.*

It is true that twinning rate varies at a large geographic scale (including as the reviewer says
being significantly higher in sub-Saharan African populations than in European populations –
see Smits and Monden 2011, cited in our manuscript). But, this observation also applies on
a much smaller scale such as between the populations we studied (Lummaa et al. 1998,
cited in our manuscript). While we do not directly test this idea (we did not collect data for
such purpose; lines 549-551), the reviewer should be pleased to learn that our new analysis
accounting for the effect of offspring mortality yield results consistent with her prediction: we
now show that variation in costs of multiple births (and not in the fertility benefits of twinning)
is sufficient to influence the strength and direction of natural selection acting on twinning
even between our similar populations (see point #4 in section "Noticeable changes to the
manuscript").

*Definition of twin mothers by first birth:*

*It is quite well known in high fertility human populations, and I think even by farmers of commercial*
*livestock for that matter, that a first birth that is twins can be very detrimental to a young mothers*
*health to the extent that it is often accompanied by infanticide or fostering (sorry no reference here).*
*So, it is possible that there could be active selection again[st] first births being twins as first births,*
*making it a slightly unreliable way to define twin mothers.*

We do show that the twinning rate was higher during first birth than for subsequent births,
which does not go in the direction suggested by the reviewer. Yet, we agree that the analysis
of the twinning status as defined by the outcome at first birth only is problematic and, as
492 explained above (see point #2 in section "Noticeable changes to the manuscript"), we have
490 now dropped this analysis from our manuscript.

*Ruth Mace*

**Reviewer #3 (Remarks to the Author):**

*This paper was a fascinating look at twinning, using robust historical datasets. The authors*
*convincingly demonstrate that we should all be studying per-birth twinning likelihood, because*
*mothers who have more births are of course more likely to have twins. They use simulations to*
*disentangle potential drivers of their observed effects. A fascinating conclusion is that the traditional*
hypothesis that some “super moms” have way more births and way more twins is not supported. This
paper was a pleasure to read, was clearly written, and raises many new research questions. Here, I
include several broad and specific comments.

*We are pleased by this very positive feedback on our work and are grateful for the thorough*
review made by this reviewer.

-----

Broad comments:

-----

**FERTILITY**

*How are you defining fertility? I have nearly always seen fertility defined as “the number of offspring*
*a mother has in her life.” You show through simulation that twinning increases the number of*
offspring, but you also show that twinning decreases birth events. It was jarring to read the following
quotes in quick succession:

• 324-6: *the total number of offspring mothers produced during their lifetime did increase with*
twinning propensity, despite the reduction that the latter imposed on total births.

• *Followed immediately by “it is twinning that impacts fertility and that such an impact is negative.”*
The way you talk about and define fertility should be carefully defined and adjusted throughout based
on conventions in the literature.

*We had defined fertility in the previous version but we agree that some statements were*
*confusing. To remedy the situation, we are now defining *intrinsic fertility* (“a woman’s*
*potential to give birth irrespective of age or past reproduction”; line 76) just before the*
*definition of *twinning propensity* (“the probability that a birth produces more than one*
*offspring”). These definitions should help the reader to understand that we deliberately want*
to exclude the effect of twinning itself from what we refer to as fertility. In most studies, it is
540 *fine for fertility to encapsulate the effect of twinning, but since the point of our study is to*
investigate the relationship between these two aspects, they cannot be intertwined. We
*thought of using an alternative term than *intrinsic fertility* but we failed to identify anything*
better. We thus did our best to rework all sentences that may have been jarring to read so as
*to avoid any ambiguity. For the same reasons, we also clarify that total births refers to the*
number of births and not offspring (lines 97-98).

**RELATEDNESS:**

• *Are you able to account at all for family structure between mothers? If impossible this should be*
noted.

We do consider that the different births within a given mother are not independent (a step
required to model maternal heterogeneity), which accounts for the largest source of
relatedness in our dataset. We did not however consider that observations from different
mothers may be dependent on their relatedness. Unfortunately, not many relatedness
studies have been published for the populations we sampled. Nonetheless, some have and
a few authors of this paper have also computed (but not yet published) the average level
relatedness in their particular dataset. The estimates obtained suggest that the average
relatedness is very low. For example, in the Krummhörn population the mean F-value
amounts to 0.00364, which corresponds to $r \sim 0.00182$ between two spouses (Johow et al.,
2019, now cited in Methods line 579).

In this light, and since our goal is not to perform a quantitative genetic analysis of twinning,
we maintain that correcting the analyses by relatedness is not justified. There is no reason to
think that the results would be noticeably altered by such a small amount of relatedness and
the computational cost implied by accounting for relatedness would be immense.

*MORTALITY of TWINS vs. SINGLETONS*

- *Something the paper is broadly missing, and that which is related to the # offspring vs. # births comment above, is the mortality rates of twins.*
- *The authors should find estimates from the human biology literature and apply them in their paper to come up with mortality-adjusted estimates of fertility (# living offspring at age 1, say) between twinning and non twinning. (or low twinning and high twinning)*
- *The non-human primate literature may be useful here, because many papers on non-human primates have documented strong effects whereby singletons have far higher survivorship than twins, a difference than arises very soon after birth (that in, twins have mortality right near birth) – see, e.g.:
? McCoy et al. (2019) "A comparative study of litter size and sex composition in a large dataset of callitrichine monkeys." American journal of primatology;
? Ward et al. (2014). Twinning and survivorship of captive common marmosets (*Callithrix jacchus*) and cotton-top tamarins (*Saguinus oedipus*). Journal of the American Association for Laboratory Animal Science;
? Harris, R. Alan, et al. (2014) "Evolutionary genetics and implications of small size and twinning in callitrichine primates." Proceedings of the National Academy of Sciences*

This is a great suggestion which has truly benefited our paper! We have now incorporated
the effect of mortality into the analysis about natural selection on twinning (see point #4 in
section "Noticeable changes to the manuscript") and cited the excellent study from McCoy et
al. 2019 (line 433; we chose to pick a single one since we have reached the total number of
citations allowed by the journal).

MATERNAL AGE

*In general, why do you not include maternal age in all of the models? It is known to strongly influence twinning rates.*

As mentioned above (see point #3 in section "Noticeable changes to the manuscript"), we
have now clarified in the text why some models include maternal age while others do not.

DATA and ANALYSIS

• It is useful in studies with lots of data to, at some point, see the real data itself rather than just the
results of models and simulations. For example, to see the distribution of # of births and # of offspring
in the data. Could this be included, and raw data plotted where possible?

We agree with the reviewer that displaying raw data is, in general, a very good idea. This is,
for example, why we had already plotted the raw distribution of the number of births in figure
S6A. The distribution of the number of offspring would look just the same on a plot since for
the great majority of women (~ 92%) the number of offspring is the same as their number of
births.

For binary variables such as the twinning status, it is however not so easy to devise useful
plots of the raw data. Another complexity when it comes to characterise the distribution of
the twinning status is that our data correspond to different (sub-)populations which show
different twinning rates and which have unequal sample sizes. Those are the reasons which
drove us to depict figures 1 & 2 as we did: those plots do not represent raw data as such, but
they are as close to them as we can do while accounting for unequal sample sizes and
expressing binary outcomes as probabilities. This is also why we had included in Table 1
and S14 columns providing the raw rate of twinning both at the level of births and mothers,
for each (sub-)population.

• In general, I think too much about the models is in the supplemental (essentially hidden from the
casual reader's view)—model predictors for key findings should be included in the figure captions
and text, such as when maternal age is considered (or not).

We have now provided more information about the models and the predictors in the main
text by introducing a new figure (new figure 3) which clarifies the model structure for each
scenario. We also modified the legends of figures 1 & 4 so as to provide more information
about the models.

EVOLUTIONARY CONTEXT OF TWINNING

*Twinning is a very cool and unusual evolutionary phenomenon, and I would like to see reference to
key*

Unfortunately the end of the sentence is missing so we can only speculate about what the
reviewer wanted. In the new version of our discussion, we now explicitly detailed the
different hypotheses about how twinning evolves and have cited the relevant papers, which
hopefully addresses the issue (lines 524-566).

-----
LINE COMMENTS
-----

59-60: repeated words

Fixed.

*Figure 1:*
• *Include statistics in the caption for the reader who likes to look at figures*

Done.

• *Include the predictors and summary stats of the GLMM models*

We have clarified which predictors are included. We did not provide summary statistics since
we would have to provide 25 different numbers and none of them would help the general
reader to interpret the plot. We have created supplementary tables for this purpose (see
Table S1 & S2 in SI). We now refer to such tables in the captions of the figures from the
main text.

• *C-D: Maternal age is a strong predictor of twinning potential. Was this included in the model of
656 twinning-at-first-birth? This could be a very simple reason why twinning at first birth decreases
lifelong births. Why not just look at per-birth twinning likelihood rather than twinning at first birth
likelihood?*

We have now removed this analysis from our manuscript (see point #2 in section
"Noticeable changes to the manuscript").

Figure 2

• *Include statistics*
• *Include summary stats and predictors of GLMM model*

Similarly to figure 1, we have clarified which predictors are included in the caption of figure 2;
we do not provide summary statistics beyond the slope of interest but again refer in the
caption to the supplementary table containing them.

• *Should the axis include 0? This is a little bit deceptive, particularly since it is already a strange axis
(although you kindly explain why in the caption)*

The y-axis cannot include 0 since the fit uses a logit transformation which transforms 0 into
minus infinity.

*151-153: This raises the question- why study first birth at all when you could instead study per birth
twinning potential? (I am suggesting you justify, rather than exclude, this analysis)? I can imagine
good reasons to do so. One might be to carefully consider maternal age without other confounding
factors such as parity. However, such a decision sort of needs to be justified given that you have a
more comprehensive approach available.*

We have now removed this analysis from our manuscript (see point #2 in section
"Noticeable changes to the manuscript").

*154-155: "this analysis confirms that the relationship between twinning
propensity and total births is not positive at the level of births."*

*I thought this sentence could be more clearly worded as something like (no need to use this exact*
*wording): “Per-birth twinning propensity is negatively related to total births. That is, mothers who*
were more likely to have twins in a given birth event were not likely to have more births overall.”

*We thank the reviewer: we have now rephrased our sentence as suggested (with minor*
*adjustment, see lines 192-193).*

*158-161: You describe the differen[tce] between a mother of 1 and mother of 18, and the confidence*
*intervals nearly overlap—could you include the same sentence for mothers + and – one or two*
*standard deviations from the median # of births so we can better understand the scale of the effect?*

*We provided estimates for the per-birth twinning probability for 1 and 18 births because the*
*relationship between total births and per-birth twinning probability is monotonic. In this*
*context, predictions associated with the minimum (i.e. 1) and maximum (i.e. 18) number of*
*births provide the full range of per-birth twinning probabilities (see Table 1). The numbers we*
*give do therefore illustrate that the variation in per-birth twinning probability is modest and*
*that twinning remains rare in all circumstances. We believe that those are important pieces*
*of information to provide.*

*We do not see what predictions associated with the median +/- 1 or 2 standard deviations*
*would achieve. Not only would it be partially redundant to the estimates provided, but*
*predictions at those values are typically associated with informal testing. Such a testing*
*procedure is valid whenever the variable is normally distributed, but this is not the case here.*
*We thus do not want to encourage such practice and instead provide confidence intervals for*
*the odds ratio which is the right information to use for anyone interested in significance*
*testing. Here, it does not include 1, so the relationship is significantly different from 0.*

*181-182: “mothers were indeed less likely to reproduce following the delivery of*
*twins than following the delivery of a singleton” Were mothers slower to have their next birth after*
*having twins? Also, what does “the probability of parity progression” mean in simple terms—the*
*likelihood that mothers have any more births, period? This should be described here and in the fig.,*
*caption.*

*We have now reworked the problematic sentence. We had already defined those*
*demographic terms in the text (now line 228-231), but we have now added the definition of*
*"parity progression" and "interbirth interval" in the caption of figure 4 (formerly figure 3) as*
*suggested by the reviewer.*

*178-206: My stylistic preference would be to avoid “mechanism A, B, C phrasing” which I have to go*
*back to refer to, and instead replace these with the actual name of the mechanisms (interbirth*
*intervals, parity, maternal age). I would prefer this change to be made throughout the whole*
*document.*

*It appears necessary to keep letters to refer to the mechanisms because our analyses*
*involve 16 combinations of up to 4 mechanisms. So referring to a given simulation scenario*
*(e.g. ACD) would be very awkward and lengthy without abbreviations. Yet, we agree with the*

734 reviewer that it is easy to lose track of what A, B, C, D refer to. We have thus now changed
the letters we used to ease recollection: A is now called P for Parity progression, B is now
called I for Interbirth intervals, C is now called S for reproductive Schedule, and D is now
called H for Heterogeneity. In short, ABCD becomes PISH (but note that we tend not to use
all the letters together as an acronym).

Another reason for sticking with abbreviations is that all mechanisms actually consider all life
history events (interbirth intervals, parity progression and twinning; see new figure 3), so
using the name of such events would be misleading.

*Figure 4:*

• *Figure 4, could you add a legend for what each letter refers to? Also, I would find it easier to parse
if it were oriented horizontally rather than in a circle, but I understand if you prefer this design
choice.*

We have now added the definition of each mechanism in the legend of figure 5 (formerly
called figure 4).

We think that our flower plot is actually easier to parse than a horizontal one (we tried both),
so we chose to keep this design.

• *It is important to note that not rejecting the null model does not mean accepting the null model*

It is correct that none of the retained scenarios may be the true one. We have now added a
sentence to express the reviewer's point (line 364-365).

208: *"heterogeneity between mothers" is ambiguous. I would suggest more precise phrasing, such as
"mothers who twin often also are more fertile overall"*

We thank the reviewer and have rephrased the sentence similarly as suggested (line 298-
300).

274-275: *"the impact of twinning events on parity progression and the effect of the schedule of
reproduction" – can this be replaced with a less jargony "parity effects and maternal age effects", or
768 is that not precise enough?*

No, we cannot rephrase as suggested since all four mechanisms considered do involve
models that include both parity and maternal age as predictors. This is now made clear
(lines 231-234, new figure 3).

274: *A simple way to understand the impact of maternal age is look only at first birth, whether or not
it is a twin, and include age as a predictor. If I am understanding your paper correctly you didn't do
that model.*

We have removed the analysis of the twinning status at first birth (see point #3 in section
"Noticeable changes to the manuscript"). As suggested by Reviewer 2, such an analysis
may reveal patterns specific to first births only.

313-327: Any conclusion about fitness drawn from this need to incorporate statistics from the
literature about twin mortality rates compared to singleton mortality rates.

This is true, we have now included mortality in our analyses (see point #4 in section
"Noticeable changes to the manuscript").

339: "misconceived study designs." Since this is the paragraph most likely to be read, after the
abstract that is, say (in a phrase) in what way the designs were misconceived.

This section has been fully rewritten.

360-366: You contrast "reduced reproductive capacity" with "foregoing further reproduction." What
actual mechanism underlies the observed "foregoing"? Is it reduced capacity? Choice? Birth
control? Reduced capacity due to a busy life caring for twins? Reduced biophysical capacity? Etc.

Yes, we referred to maternal choice. We have now clarified the sentence (line 254).

368: Here, the reader has forgotten what Mechanism A is (if they are tired and forgetful like me, at
least).

We no longer refer to mechanisms by letters within the discussion.

403: between populations? What between-population analyses did you do?

The analysis testing for an interaction between total birth and population was shown in
Methods but we have now moved it to Results (lines 195-198) and written a new paragraph
in methods to provide more details about the underlying analysis (lines 674-687).

406: You have not adjusted for mortality in order to conclude: "the additional children brought by a
twin birth more than compensated for the reproductive cost that twinning imposes on mothers."

This is correct, but we have now revised this statement based on our new results
considering differences in mortality (see point #4 in section "Noticeable changes to the
manuscript").

414-422: including much recent work on primates, including:

We have rephrased accordingly and cited the study by McCoy et al. 2019 as mentioned
above.

423-462: great explanation!

We thank the reviewer for this compliment. The explanation in question has now been
moved to the introduction (lines 109-136).

476-482: *can you cite some of the studies I question here, both to demonstrate that it is a real
problem and draw attention to this issue?*

Examples of the studies in question were cited in the previous paragraph. We did not cite
them at the referred location because until those studies are being replicated using the
832 correct methodological approach, no one can tell if a given published result is correct or not.
We have now explained this in the text (lines 559-562).

834

800-802: *wonderful to have the R Package available.*

836

We thank the reviewer for this compliment.

REVIEWER COMMENTS

Reviewer #1 (Remarks to the Author):

I appreciate the authors' efforts to address the reviewers' concerns and the opportunity to see this manuscript again; this version is an improvement over the previous draft. The strengths of the manuscript are: (1) it draws attention to the data aggregation problem with a clear application to the relationship between number of births and twinning, and in doing so casts doubt on studies that assume, or have found, a link between twinning and fertility; (2) it adds to the understanding of how selection can act on human reproductive strategies through dizygotic twinning (note that I didn't say the reproductive strategy of twinning (see below)). For example, the evidence provided in Figs 1 and 2 that mothers who produce twins have more births when the data are aggregated, but fewer when the increased opportunity to produce twins is controlled, is an excellent illustration of the ecological fallacy. Likewise, the finding (Fig 4A) that producing twins entails a cost in that after producing twins women are less likely to proceed to a subsequent parity is, to my knowledge, an unrecognized fitness cost associated with twinning/double ovulation. The finding that after producing twins, women have slightly shorter interbirth intervals than women who have birthed singletons (Fig 4B) is not surprising given that the higher neo- and postnatal mortality of twins could cause mothers to return to ovulation sooner than mothers of singletons. Similarly, that the probability of parity progression decreases with maternal age is not surprising, being consistent with the well documented decrease in the probability of live birth per zygote as women age. However, the difference in parity progression probability for twinning and non-twinning moms is opposite that found by Robson and Smith, which should be discussed.

The manuscript could be improved if the authors showed more care in how they define intrinsic fertility. In the abstract it is the predisposition to conceive (i.e. become pregnant); in the introduction it is the propensity to give birth irrespective of age or past reproduction. These definitions are not the same, as it is well documented that the predisposition to conceive (abstract definition) decreases with maternal age, but the definition in the introduction as propensity to give birth irrespective of age contradicts this. So, in contrast to probability to live birth per zygote, which is clearly related to twinning (see below), I don't know what intrinsic fertility is relative to how it might be related to twinning. (If one Googles intrinsic fertility, the definition that appears is from the artificial reproductive technology literature as the probability of live birth per oocyte retrieved.) Operationally, the authors note that intrinsic fertility has been measured as lifetime number of births (lines 95-103), and this is what leads to the ecological fallacy, which concerns the correlation between number of births and twinning, a strong point of the paper, as the authors show that when risk of twinning is considered, twinning mothers have fewer births (see above). Therefore, I think it is critically important that the authors take care that they are consistent in what they mean by intrinsic fertility.

The results in Fig 4C are interesting for two reasons. The first is that the lack of an increase in elevation of the twinning rate function for women with increasing parities is contrary to that reported by Bulmer in his book on twinning, which the authors cite in other contexts. This difference should be discussed. The second reason gets at the secondary focus of the manuscript, the optimal twinning rate and differences between populations in twinning rates, which I feel misses the mark.

The shape of the twinning rate function on maternal age in Fig 4C is a direct consequence of two underlying functions, the probability of live birth per zygote on maternal age and the probability of double ovulation on maternal age (see Atkinson's 1985 formula and Fig 1 in Hazel et al. 2020). Ignoring triple and higher-level ovulations because of their rarity, a woman can only produce twins if she double ovulates and both embryos survive to live birth. The probability of this happening can easily be calculated from the probability of double ovulation per cycle and the per zygote probability of survival to birth (Atkinson's 1985). Twinning is therefore a complex trait, reflecting two separate traits, double ovulation and prenatal survival from fertilization to birth, both of which are dependent on age. This age dependence adds to the complexity of comparisons of twinning rates between populations.

That twinning can only occur if double ovulation occurs begs the question of whether it is even worthwhile to talk about an optimal twinning rate (line 546). This is especially true considering recent

simulation results (Hazel et al. 2020) which showed that, given the well documented decline in probability of live birth per zygote with increasing maternal age and the equally well documented costs of producing twins (reduced maternal and offspring survival), an age dependent double ovulation strategy was superior to an always single or always double ovulation strategy--but only when double ovulation could result in twins. If women that would normally produce twins could abort one of the two, then an obligate double ovulation strategy was most successful. That is, the optimal twinning rate was zero, while the optimum double ovulation rate was 100%. This is clear evidence for the hypothesis that twins are a byproduct of double ovulation. Likewise, "the clear peak in twinning probability for women in their mid to late thirties" depicted in Fig 4C is more quantitative than "qualitative" evidence for the ova insurance hypothesis (see discussion lines 524-542), since that peak can only happen if the probability of double ovulation increases with age as the probability of live birth per zygote falls. Therefore, a more realistic way to think of twinning rates is via their contribution to how selection molds an optimal double ovulation rate that is dependent on age (given that the optimum strategy of double ovulating but not producing twins does not appear to be physiologically possible). Along these lines, perhaps the reason the results presented "casts doubt on the validity of clinical and epidemiological studies that assumed that the lifetime (dizygotic) twinning status is a proxy for female fertility" (lines 480-481) is because double ovulation is more strongly tied to number of births than is twinning. This is because only a fraction of the women who double ovulate produce twins. For example, in European populations, where the probability of survival to birth is estimated at about 20% for women in their mid to late twenties (see Fig 1f in Hazel et al. 2020), only 4% of double ovulations will produce twins. Because the evidence suggests that double ovulation is increasingly likely in older women, when prenatal survival is low, the link between twinning and fertility will indeed be slight.

The simulation results in this manuscript, while interesting, do not in my opinion significantly add to the understanding of twinning. For example, the simulations reported in Hazel et al 2020 simulated the reproductive lives of women from menarche to menopause, following zygotes from each ovulation in double ovulating and single ovulating women until the offspring reached age 15, and estimated the lifetime reproductive success of women switching from single to double ovulation at different ages, or always single ovulating or always double ovulating. Those simulations were able to capture the effects of both prenatal and postnatal mortality of offspring, and mortality differences of mothers birthing twins and singletons on the optimal age of switching to double ovulation. The simulations produced estimates of age dependent costs and benefits of double and single ovulation and how the production of twins influenced those costs. In contrast, the simulations in this manuscript, by concentrating only on twinning, fail to capture the full cost and benefits of single versus double ovulation, without which twinning could not occur.

To summarize, the manuscript's principal contribution is in how it highlights the problem of data aggregation in an interesting application to studies examining the relationship between twinning and intrinsic fertility. But the definition of intrinsic fertility is unclear. The study contributes to the study of human reproductive life history traits, but the concentration on twinning as what is optimized is misplaced. The strategy on which selection must act for twinning to exist is whether women should ovulate one or two ova, and when they should do so. The effect of producing twins on reproductive success is an important part of that story, but not as important as the authors would have us believe. I do hope a revised version of this manuscript is published somewhere because the results do provide some new information. However, for it be accepted for publication by such a prestigious journal as Nature Communications the authors need to address how their findings significantly add to the understanding of twinning beyond that which was gained by Hazel et al. 2020.

Reviewer #2 (Remarks to the Author):

I think the comments have been adequately addressed I am happy to accept.

Reviewer #3 (Remarks to the Author):

I am happy with the thorough revisions which, in my opinion, have addressed all reviewer comments. The article is a pleasure to read and a significant contribution.

Point-by-point response to the reviewers' comments

REVIEWER COMMENTS

Reviewer #1 (Remarks to the Author):

I appreciate the authors' efforts to address the reviewers' concerns and the opportunity to see this manuscript again; this version is an improvement over the previous draft. The strengths of the manuscript are: (1) it draws attention to the data aggregation problem with a clear application to the relationship between number of births and twinning, and in doing so casts doubt on studies that assume, or have found, a link between twinning and fertility; (2) it adds to the understanding of how selection can act on human reproductive strategies through dizygotic twinning (note that I didn't say the reproductive strategy of twinning (see below)). For example, the evidence provided in Figs 1 and 2 that mothers who produce twins have more births when the data are aggregated, but fewer when the increased opportunity to produce twins is controlled, is an excellent illustration of the ecological fallacy. Likewise, the finding (Fig 4A) that producing twins entails a cost in that after producing twins women are less likely to proceed to a subsequent parity is, to my knowledge, an unrecognized fitness cost associated with twinning/double ovulation.

We are very happy to read that the reviewer sees our manuscript improved, presenting several strengths, as well as novel results.

The finding that after producing twins, women have slightly shorter interbirth intervals than women who have birthed singletons (Fig 4B) is not surprising given that the higher neo- and postnatal mortality of twins could cause mothers to return to ovulation sooner than mothers of singletons.

The reviewer is correct that observing a reduction in the interbirth interval after a twinning event is not particularly surprising in the context of the high neo- and postnatal mortality, and we make this point in the paper by citing previous studies that have described this, e.g. "*the higher mortality of twin offspring (Bulmer, 1970; Kleinman et al. 1991; Gabler and Volland, 1994; Sear et al., 2001; Smith et al., 2014; van Heesch et al., 2015; Monden & Smits, 2017) may lead to shorter interbirth intervals.*" (lines 273-274).

Similarly, that the probability of parity progression decreases with maternal age is not surprising, being consistent with the well documented decrease in the probability of live birth per zygote as women age.

We agree that we should have made this clearer. We have therefore modified a sentence and added a key reference on this topic (line 265-266, ref = Broekmans et al., 2009).

However, the difference in parity progression probability for twinning and non-twinning moms is opposite that found by Robson and Smith, which should be discussed.

Contrary to the analysis of parity progression by Robson & Smith (2012 [cited in main text]), our Fig 4A does not illustrate how the parity progression differed between twinning and non-twinners. Instead, it illustrates how parity progression differed after a twin vs a singleton birth.

The difference is subtle but of importance. Indeed, as we emphasise repeatedly in our paper, the research design used by Robson & Smith (2011, 2012 [both cited]) can generate misleading results because it relies on the analysis of aggregated data (e.g. lines 95-136). Here, in particular, any variation in total birth can generate an association between twinning and parity progression that is independent of any possible variation in twinning caused by risk factors. This is because while an increase in total birth necessarily implies an increase in parity progression probabilities, it also increases the exposure to the risk of twinning and thus the probability for mothers to be classified as twinning when data are aggregated. Because of this drawback, we cannot tell if the differences observed between our results and those from Robson & Smith stem from a genuine biological difference, a methodological difference, or both.

In this light, we concluded that comparing our results on parity progression to those from Robson & Smith 2012 would not be appropriate. The only modification that seemed necessary for us to address this comment was to add an additional reference to Robson & Smith 2012 (lines 527), which makes it explicit that this study also suffers from the general issue we have described in details for Robson & Smith 2011 and other studies.

[Technical note: the fact that in Robson & Smith (2012) the comparison between twinning and non-twinners for parity progression is performed for each parity independently does not circumvent the aforementioned issue. Indeed, such a standardisation procedure does not eliminate the effect of variation in the exposure to the risk of twinning. For this, one would either need to compare mothers with the same total exposure (i.e. same total birth) at any given parity, or, as we did, one would need to analyse (non-aggregated) birth-level data.]

The manuscript could be improved if the authors showed more care in how they define intrinsic fertility. In the abstract it is the predisposition to conceive (i.e. become pregnant); in the introduction it is the propensity to give birth irrespective of age or past reproduction. These definitions are not the same, as it is well documented that the predisposition to conceive (abstract definition) decreases with maternal age, but the definition in the introduction as propensity to give birth irrespective of age contradicts this. So, in contrast to probability to live birth per zygote, which is clearly related to twinning (see below), I don't know what intrinsic fertility is relative to how it might be related to twinning. (If one Googles intrinsic fertility, the definition that appears is from the artificial reproductive technology literature as the probability of live birth per oocyte retrieved.) Operationally, the authors note that intrinsic fertility has been measured as lifetime number of births (lines 95-103), and this is what leads to the ecological fallacy, which concerns the correlation between number of births and twinning, a strong point of the paper, as the authors show that when risk of twinning is considered, twinning mothers have

fewer births (see above). Therefore, I think it is critically important that the authors take care that they are consistent in what they mean by intrinsic fertility.

It is correct that we coined the term ***intrinsic fertility*** (defined lines 75-77). The reason for this new term is that there is no established term used to distinguish explicitly the variation in fertility that happens as the result of age, parity, and stochastic events within the lives of mothers, from the variation that is independent of these factors and which thus emerges as the result of fixed (i.e. intrinsic) differences in fertility between women.

Within papers that are close to our work, we only noticed a few instances of authors attempting to make an explicit distinction between the different types of fertility. In particular, Sear et al. (2001 [cited]) simply used "***fertility***" as a synonym of what we call "***intrinsic fertility***". Thus, to make the distinction between intrinsic fertility and the fertility that varies with age, they called the latter "***age-specific fertility***". For our paper this term cannot be used because it would not capture other sources of variation within the life of a mother, such as the parity number. When Robson & Smith (2011 [cited]) related Sear et al.'s results on intrinsic fertility, the authors opted for another term – "***overall fertility***". Again, the term is not very precise and could be confusing. It is also not defined in the paper, although the authors connect it to the notion of heterogeneity stemming from "***subjects [that] differ in their inherent quality***". Again, they defined neither "***inherent***" nor "***quality***" but in context, we interpret "***inherent***" as a synonym for what we called "***intrinsic***".

Within biology at large, the most articulated discussions about the terminology used around sources of heterogeneity that connect to our topic can be found in evolutionary demography. In a seminal paper, Tuljapurkar et al. 2009 [Tuljapurkar, S., Steiner, U. K., & Orzack, S. H. (2009). Dynamic heterogeneity in life histories. *Ecol. Lett.*, 12(1), 93–106.] defined "***fixed heterogeneity***" as the "differences between individuals that are fixed at birth", which they contrasted with "***dynamic heterogeneity***" arising "***when stage transitions are probabilistic, and different individuals may follow different sequences of stages as they age***". There is however no need to consider that birth must be the defining moment where the differences in focus are set and Cam et al. 2016 [Cam, E., Aubry, L. M., & Authier, M. (2016). The conundrum of heterogeneities in life history studies. *Trends Ecol. Evol.*, 31(11), 872–886.] thus proposed a more encompassing terminology. Their "***Hidden Persistent Demographic Heterogeneity (HPDH)***" describes unobserved individual characteristics that are fixed after individuals entered the study". The terminology is general, not focussed on a particular trait, but fits tightly with the idea we are trying to communicate. (In our case, individuals would be considered as entering the study when they had their first birth.)

This study of the literature left us with the following options for our term: intrinsic fertility, overall fertility, inherent fertility, fixed fertility, or hidden persistent fertility. We looked at other works but failed to identify better alternatives than those aforementioned. One possible exception may have been ***potential fertility***, which JM Gaillard suggested to us; but we noticed that the term is already widely used to refer to aspects of fertility that actually vary within individuals, so that disqualifies this last proposition. We prefer intrinsic and inherent over the alternatives which

have broader meanings and could thus more easily be misunderstood. Between intrinsic and inherent, we picked intrinsic although we have no clear preference – they are considered synonymous in modern English and their etymological roots are also similar.

These considerations reflect that we agree with the reviewer that care must be taken when writing around the different facets of fertility, but the works mentioned above also show it is a delicate matter. The two locations where mentioning intrinsic fertility remains most difficult are the title and the abstract. At these early stages, the readers may neither know the term, nor anticipate the intended meaning. For the title we opted to use "*fertility*" without any qualifier because the title encapsulates both our findings about "*intrinsic fertility*" (as shown by the negative correlation in random effects) and our findings about "*realised fertility*" – the actual number of births women experienced (simulation results). The usage of "*fertility*" without qualifier thus fits nicely here since what we wrote is true for the two aspects of fertility that can be defined at the level of mothers. For the abstract, we did use the term *intrinsic fertility* and defined it. Yet, due to the drastic word count limitation (150 words), we opted for something more simple and shorter than the proper definition given in the introduction. We thus defined intrinsic fertility as "*a physiological predisposition to conceive easily*", which we aimed to be perceived as synonymous to the full definition ("*a woman's potential to give birth irrespective of age or past reproduction*"). This is why we chose the word *predisposition* ("*to dispose in advance*"; Merriam-webster dictionary) and not *disposition* ("*the tendency of something to act in a certain manner under given circumstances*"). That the reviewer perceived the two definitions are very different, suggests however that we failed to achieve our intended goal. We have thus now revised our abstract where we now define intrinsic fertility as "*a tendency to conceive easily irrespective of age and other factors*". Everywhere else in the text we either used *intrinsic fertility* or dropped the qualifier *intrinsic* when we talk about fertility in general. We took the opportunity of these minor revisions to refine the full definition (now "*potential to give birth irrespective of age and any stochastic factors occurring within her reproductive life, including past reproduction*", lines 75-77) double check the usage of all 71 mentions of fertility in our paper. This led to us to rework a few sentences for clarity throughout.

The results in Fig 4C are interesting for two reasons. The first is that the lack of an increase in elevation of the twinning rate function for women with increasing parities is contrary to that reported by Bulmer in his book on twinning, which the authors cite in other contexts. This difference should be discussed.

The reported effect of parity upon the per-birth probability of twinning appears highly variable across studies. We now mention this in the text (lines 295-300). As observed by the reviewers, a few previous studies have concluded that twinning increased with parity. Among such studies, some did not control for maternal age (e.g. Nigeria: Nylander 1981 [cited]; Sweden: Lichtenstein et al. 1996 [cited]) and thus these results could be driven by the influence of maternal age alone. Yet, others also reported such a positive effect after standardising twinning rate by maternal age (e.g. Italy: Bulmer 1970 [cited]; US: Allen & Parisi 1990 [now cited lines 297-298]). Some studies do report, however, results similar to ours (a higher twinning rate at first parity); this is, for example, the case of Obi-Osius et al., 2004 [now cited line 295] who studied a

German population. Moreover, the majority of studies we looked at failed to detect any relationship between twinning and parity (e.g. Denmark: Bønnelykke 1990 [Bønnelykke, B. (1990). Maternal age and parity as predictors of human twinning. *Acta Genet. Med. Gemellol.*, 39(3), 329–334.]; Germany: Gabler & Volland 1994 [cited]; Gambia: Sear et al. 2001 [cited]; Denmark: Morales-Suárez-Varela et al. 2007 [Morales-Suárez-Varela, M. M., Bech, B. H., Christensen, K., & Olsen, J. (2007). Coffee and smoking as risk factors of twin pregnancies: The Danish National Birth Cohort. *Twin Res. Hum. Genet.*, 10(4), 597–603.]; Norway: Skjærvø et al. 2009 [cited]).

Whether the variation in the documented effect of parity reflects biological differences between populations is unknown. Even if we suspect some differences between populations, it is not possible to easily assess this hypothesis because one would have to apply the same statistical methods to all populations compared. This would be all the more important as the age and parity predictors covary (here presenting a correlation of ca. $\rho = 0.69$; now mentioned line 290) and their effects are not linear (for figure 4C they are best fitted by a polynomial of order 3 in both age and parity). In such cases, the different models used in the literature to estimate the effect of age and parity could easily lead to opposite conclusions. This has been particularly highlighted when at least one of the predictors has weak effect (which is the case here for parity) (Mason et al. 1991 [now cited line 753]). In our analyses, we minimised the risk of incorrect inference by selecting the best-fitting order of the polynomial, and a similar procedure should be applied to other data to allow a formal assessment of heterogeneity between populations.

Importantly, in all analyses other than that shown in Fig 4 we considered the joint effect of parity and maternal age together without trying to split the effect of each variable into different statistical models (see Fig 3). We did this precisely because we recognised that collinearity could be a problem otherwise. Our approach thus made sure to circumvent the issue when comparing the different mechanisms that may drive the relationship between twinning and fertility. This is now explicitly mentioned in lines 752-757.

The second reason gets at the secondary focus of the manuscript, the optimal twinning rate and differences between populations in twinning rates, which I feel misses the mark.

The shape of the twinning rate function on maternal age in Fig 4C is a direct consequence of two underlying functions, the probability of live birth per zygote on maternal age and the probability of double ovulation on maternal age (see Atkinson's 1985 formula and Fig 1 in Hazel et al. 2020). Ignoring triple and higher-level ovulations because of their rarity, a woman can only produce twins if she double ovulates and both embryos survive to live birth. The probability of this happening can easily be calculated from the probability of double ovulation per cycle and the per zygote probability of survival to birth (Atkinson's 1985). Twinning is therefore a complex trait, reflecting two separate traits, double ovulation and prenatal survival from fertilization to birth, both of which are dependent on age. This age dependence adds to the complexity of comparisons of twinning rates between populations.

This is correct, and we did mention both double ovulation and prenatal survival in our text as responsible for the pattern shown in Fig 4C. Here is what we wrote: "*we observed a clear peak in twinning probability for women in their mid to late thirties. This particular pattern (see also Bulmer, 1970; Nylander, 1981; Gabler & Volland, 1994; Lichtenstein et al., 1996; Hazel et al., 2020) is (qualitatively) predicted by the ova insurance hypothesis (Anderson, 1990; Hazel et al., 2020). This hypothesis states that dizygotic twinning occurs as a by-product of polyovulation, a condition-dependent compensatory mechanism against embryo mortality selected to increase with maternal age. It predicts women reproducing early in their life will tend to have more singletons because polyovulation is rare, and women reproducing late will tend to have more singletons because their polyovulation is masked by the high rate of embryo mortality (Hazel et al., 2020).*" (lines 542–551).

That twinning can only occur if double ovulation occurs begs the question of whether it is even worthwhile to talk about an optimal twinning rate (line 546). This is especially true considering recent simulation results (Hazel et al. 2020) which showed that, given the well documented decline in probability of live birth per zygote with increasing maternal age and the equally well documented costs of producing twins (reduced maternal and offspring survival), an age dependent double ovulation strategy was superior to an always single or always double ovulation strategy--but only when double ovulation could result in twins. If women that would normally produce twins could abort one of the two, then an obligate double ovulation strategy was most successful. That is, the optimal twinning rate was zero, while the optimum double ovulation rate was 100%. This is clear evidence for the hypothesis that twins are a byproduct of double ovulation.

This is an interesting remark but unfortunately the results mentioned by the reviewer are far from providing "*clear evidence for the hypothesis that twins are [only] a byproduct of double ovulation*". Indeed, as for any simulation work, Hazel et al. (2020 [cited])'s results rely on several key assumptions. Of particular importance, the only cost they considered in their simulations which limit the evolution of constant double ovulation strategy (called "*double ovulators*") is twinning. Females are assumed to have an unlimited amount of ova until they reach a fixed age of 40 years. Thus in the absence of any twinning, it is not surprising that double ovulators show an average fitness higher than conditional strategists (mothers whose double ovulation only starts after a given age). In fact, based on their assumptions, they could not have obtained a different result. That the conditional strategy is best "*only when double ovulation could result in twins*" is just the direct consequence of modelling double ovulation limiting the effect of reproductive senescence without considering any cost acting on double ovulation per se. Had Hazel et al. considered that mothers may run out of ova, or that there would be any physiological cost associated with double ovulation, their results may have been very different. Further, within their model, twinning persists only by constraints on possible phenotypes (if a mutant suppressed all twinning by enforcing death of one of the two zygotes on their way to become twins, it would be favoured by selection). In this perspective, there is no "optimal twinning rate" because such a strategy is assumed impossible. But no evidence is presented for

this assumption, and we feel it legitimate to consider other assumptions for which the concept of optimal twinning rate is more meaningful.

This is why in this part of the discussion, we examine how our results weigh on the *different* existing hypotheses proposed to account for the origin and maintenance of twinning in humans. We agree that the hypothesis proposing that double ovulation is the primary target of selection seems very likely to be correct. Yet, we recall that there is no direct support for it, only indirect evidence which we list (lines 543 + 551–555). It thus appears legitimate not to disregard other hypotheses. We have now clarified this (lines 568–576).

Likewise, “the clear peak in twinning probability for women in their mid to late thirties” depicted in Fig 4C is more quantitative than “qualitative” evidence for the ova insurance hypothesis (see discussion lines 524-542), since that peak can only happen if the probability of double ovulation increases with age as the probability of live birth per zygote falls.

We disagree with the reviewer. The evidence that Hazel et al. provided is qualitative and not quantitative. This is because their conceptual model predicts that twinning will peak with maternal age, and it does not predict when that peak occurs (the quantitative elements). It is true that the illustrations they provide are quantitative; yet, a careful reading of their methods clearly shows that the parameters they fit are precisely fitted so that their peaks fit best the patterns observed in real data.

Therefore, a more realistic way to think of twinning rates is via their contribution to how selection molds an optimal double ovulation rate that is dependent on age (given that the optimum strategy of double ovulating but not producing twins does not appear to be physiologically possible).

To clarify one point before answering, the reviewer should have written "*given that the optimum strategy of double ovulating but not producing [any] twins does not appear to be physiologically possible*". Indeed, as the reviewer is certainly well aware, a very large fraction of twins formed following double ovulation lead to the birth of a singleton precisely because one of the two eggs is reabsorbed by the mother – an event referred to as "*vanishing twin*" (see e.g. Hall 2003 [cited]).

We agree with the reviewer that "*how selection molds an optimal double ovulation rate that is dependent on age*" is an interesting question. We fear however that to precisely model the *moulding*, one would need much better knowledge about the true age-specific patterns of double ovulations and prenatal loss. For now, there is hardly any actual data on those which is why neither Hazel et al. nor ourselves attempted this exercise and instead considered a very subjective mathematical formulation of the process when investigating how different reproductive strategies impact both twinning and women's realised fertility.

To better understand how the subjective mathematical formulations of the process differ between Hazel et al. and our paper, it is important to recall how the two pieces of work model the relationship between age and double ovulation.

In Hazel et al.'s work, the modelling of the age-specific double ovulation is explicit. The authors modelled the effect of age on double ovulation by a simple threshold: before a given age, women release one ovum per cycle, after that age, they release two. Yet, since the age used for the threshold is drawn from a random distribution and since many women are considered, the resulting effect is that the frequency of double ovulation increases with age as a cumulative normal function (see Extended Data Fig 1 in their paper). Holding other assumptions of the model constant, it would be mathematically equivalent to see their model as representing an average woman whose probability to double ovulate would increase with age as a cumulative normal function too. To model alternative reproductive strategies, the authors considered either this cumulative normal function (conditional strategists) or that mothers always (double ovulators) or never (single ovulators) performed double ovulation. Importantly, the authors did not compare the fitness of alternative conditional strategists differing in their exact age-specific double ovulation.

In our paper, we modelled age-specific double ovulation more implicitly since we directly modelled the age-specific probability of twinning. Yet, under the light of the ova insurance hypothesis, this can be seen as modelling the combined effect of double ovulation and prenatal loss across ages. Since our statistical parameterisation is particularly flexible, it could thus closely approximate a double ovulation that increases with age as a cumulative normal function (as in Hazel et al.), as well as many alternative forms. When we compared different reproductive strategies, we did not compare single ovulators, conditional strategists and double ovulators, but rather different conditional strategists. For us, this made more sense since it seems safe to assume that in nature all women are conditional strategists and thus that only differences among this group may explain differences in twinning rate within and among populations.

To be more specific, what we did in our simulation approach on "*Twinning and total number of offspring*" was to explore the consequences of a change in twinning propensity upon womens' reproductive success by changing the intercept of model "12" (from the Supplementary Information: formula = $T \sim 1 + \text{poly}(\text{cbind}(\text{age}, \text{parity}), 3) + (1|\text{pop})$). At the biological level, such a change can thus be understood as the overall outcome of an increase in the probability of double ovulation and a decrease in the probability of prenatal loss at all ages. If one is willing to assume that the age-specific decrease in the probability of prenatal loss is fixed within a population (as in Hazel et al.), then our simulation experiments would precisely capture the effect of increasing double ovulation. In such a case, the increase in double ovulation relative to the baseline age-specific value would apply to any age. Interestingly, within the range of parameters used by Hazel et al., this is exactly what would happen in their study if they reduced the age threshold at which double ovulation starts. One can easily check in that in R as follows:

```
f1 <- function(age) pnorm(age, mean = 41.91, sd = 9.46) # using mean estimates in Table 1 from Hazel et al.
f2 <- function(age) pnorm(age, mean = 41.91 - 1, sd = 9.46) # simulating increase in double ovulation
curve(f1, from = 15, to = 40, log = "y")
curve(f2, from = 15, to = 40, col = "red", add = TRUE, log = "y") # the increase applies to all ages!
```

In sum, while our modelling of double ovulation is more implicit than the one from Hazel et al., it is fully compatible with it.

Along these lines, perhaps the reason the results presented “casts doubt on the validity of clinical and epidemiological studies that assumed that the lifetime (dizygotic) twinning status is a proxy for female fertility” (lines 480-481) is because double ovulation is more strongly tied to number of births than is twinning. This is because only a fraction of the women who double ovulate produce twins. For example, in European populations, where the probability of survival to birth is estimated at about 20% for women in their mid to late twenties (see Fig 1f in Hazel et al. 2020), only 4% of double ovulations will produce twins. Because the evidence suggests that double ovulation is increasingly likely in older women, when prenatal survival is low, the link between twinning and fertility will indeed be slight.

Yes, it is true that there is no reason to expect a strong relationship between fertility and twinning in the light of the ova-insurance hypothesis. Yet many people misconceived such a relationship as strong due to the effect of aggregated data. We have now added a few sentences reflecting on this idea in our manuscript (lines 568-576).

The simulation results in this manuscript, while interesting, do not in my opinion significantly add to the understanding of twinning. For example, the simulations reported in Hazel et al 2020 simulated the reproductive lives of women from menarche to menopause, following zygotes from each ovulation in double ovulating and single ovulating women until the offspring reached age 15, and estimated the lifetime reproductive success of women switching from single to double ovulation at different ages, or always single ovulating or always double ovulating. Those simulations were able to capture the effects of both prenatal and postnatal mortality of offspring, and mortality differences of mothers birthing twins and singletons on the optimal age of switching to double ovulation. The simulations produced estimates of age dependent costs and benefits of double and single ovulation and how the production of twins influenced those costs. In contrast, the simulations in this manuscript, by concentrating only on twinning, fail to capture the full cost and benefits of single versus double ovulation, without which twinning could not occur.

We agree that the study from Hazel et al. is excellent and that the simulation results they present are very relevant when it comes to explaining the relationship between maternal age and twinning rate. We give due credit to this study in our text and cite this study repeatedly. We disagree however that we failed to “capture the full cost and benefits of single versus double ovulation”. As discussed above, we modelled double ovulation implicitly in a way that is fully consistent with Hazel et al.’s formalism. We did consider the cost of twinning explicitly and the increasing cost of prenatal death with maternal age is implicitly modelled by our polynomial age effect, similarly to what we explained above for double ovulation. In fact, we don’t see any costs that Hazel et al. considered that are not accounted for by our model. Since their modelling is more proximate than ours, they have to be more explicit about how the costs vary with age, which presents pros (it is explicit) and cons (results are shaped by the specific (unknown) form that must be assumed for these costs). Depending on what the goal of the study is, choosing to

model a particular component in an explicit manner or not may be favoured. This brings us to our main point: the goal of our study is different from that of Hazel et al. In their own words, Hazel et al. "*have only attempted to explain the evolution of age-dependent double ovulation*". In contrast, our study aims at explaining the relationship between twinning and fertility and its evolutionary consequences. We thus see our two studies as complementary and consider them both as important for anyone interested in the causes and consequences of twinning.

To summarize, the manuscript's principal contribution is in how it highlights the problem of data aggregation in an interesting application to studies examining the relationship between twinning and intrinsic fertility. But the definition of intrinsic fertility is unclear. The study contributes to the study of human reproductive life history traits, but the concentration on twinning as what is optimized is misplaced. The strategy on which selection must act for twinning to exist is whether women should ovulate one or two ova, and when they should do so. The effect of producing twins on reproductive success is an important part of that story, but not as important as the authors would have us believe. I do hope a revised version of this manuscript is published somewhere because the results do provide some new information. However, for it be accepted for publication by such a prestigious journal as Nature Communications the authors need to address how their findings significantly add to the understanding of twinning beyond that which was gained by Hazel et al. 2020.

We hope that the new modifications we made to our paper and the detailed responses we gave above will satisfy the reviewer and help her/him/them understand why our study brings new results, interesting in their own right.

Reviewer #2 (Remarks to the Author):

I think the comments have been adequately addressed I am happy to accept.

We thank the reviewer.

Reviewer #3 (Remarks to the Author):

I am happy with the thorough revisions which, in my opinion, have addressed all reviewer comments. The article is a pleasure to read and a significant contribution.

We thank the reviewer.